# Glycoengineered keratinocyte library reveals essential functions of specific glycans for all stages of HSV-1 infection

Ieva Bagdonaite [1] ✉, Irina N. Marinova [1], Asha M. Rudjord-Levann [1], Emil M. H. Pallesen[1], Sarah L. King-Smith[1], Richard Karlsson [1], Troels B. Rømer [1], Yen-Hsi Chen[1], Rebecca L. Miller [1], Sigvard Olofsson[2], Rickard Nordén [2], Tomas Bergström[2], Sally Dabelsteen [3] & Hans H. Wandall [1] ✉

Viral and host glycans represent an understudied aspect of host-pathogen interactions, despite potential implications for treatment of viral infections. This is due to lack of easily accessible tools for analyzing glycan function in a meaningful context. Here we generate a glycoengineered keratinocyte library delineating human glycosylation pathways to uncover roles of specific glycans at different stages of herpes simplex virus type 1 (HSV-1) infectious cycle. We show the importance of cellular glycosaminoglycans and glycosphingolipids for HSV-1 attachment, N-glycans for entry and spread, and O-glycans for propagation. While altered virion surface structures have minimal effects on the early interactions with wild type cells, mutation of specific O-glycosylation sites affects glycoprotein surface expression and function. In conclusion, the data demonstrates the importance of specific glycans in a clinically relevant human model of HSV-1 infection and highlights the utility of genetic engineering to elucidate the roles of specific viral and cellular carbohydrate structures.

Upon infection of human cells with enveloped viruses, the virus hijacks the secretory pathway leading to the decoration of the newly formed viral proteins and membranes with host glycans. The resulting glycans influence several aspects of viral biology, with the most appreciated function being immunological shields that prevent the immune system from recognizing the viral particles[1]. However, other functions of the abundantly present cellular and viral glycans are still underexplored mainly because of the lack of accessible tools and model systems.

Glycosylation is among the most diverse post-translational modifications, dynamically shaping the glycans on proteins and lipids traveling through the secretory pathway, and comprising an extensive glycocalyx on the cell surface. There are three major classes of cell surface glycoconjugates—glycoproteins, glycosphingolipids (GSLs)

and proteoglycans carrying long glycosaminoglycan (GAG) chains—that all execute numerous functions, including fine-tuning of molecular interactions, intra- and intercellular communications, and immune modulation[2–4]. Due to non-template driven synthesis and the resulting heterogeneity it is methodologically challenging to investigate the functions of specific glycans in biological systems. We have recently generated glycoengineered cell libraries by CRISPR/Cas9 gene editing composed of cells lacking different glycosyltransferases delineating human glycosylation pathways to dissect the importance of glycosylation in cell and molecular biology[5–8]. Such libraries thus represent an appealing platform for probing host-pathogen interplay.

Herpes simplex virus type 1 (HSV-1) is a widespread human pathogen persisting in sensory neurons for a lifetime and causing

[1]Copenhagen Center for Glycomics, Institute of Cellular and Molecular Medicine, University of Copenhagen, DK-2200 Copenhagen, Denmark. [2]Department of Infectious Diseases, Institute of Biomedicine, University of Gothenburg, SE-41346 Gothenburg, Sweden. [3]Department of Odontology, University of Copenhagen, DK-2200 Copenhagen, Denmark. ✉e-mail: ieva@sund.ku.dk; hhw@sund.ku.dk

recurrent cutaneous lesions due to reactivation[9]. Importantly, it can cause life-threatening CNS complications, which is a major risk for both immunocompetent and immunocompromised individuals, and may predispose to neurodegenerative disease[10-12]. HSV-1 is a large dsDNA enveloped virus, encoding 12 surface glycoproteins important for viral entry, particle formation, and cell-to-cell spread[13,14]. HSV-1 glycoproteins are confirmed to acquire both N-linked and mucin type O-linked glycans, and we have previously demonstrated the importance of elongated O-linked glycans for HSV-1 propagation[15-20]. On the host side, HSV-1 interacts with several cellular receptors including Nectin 1, herpes virus entry mediator (HVEM), 3-O-sulfated heparan sulfate, integrins, and paired immunoglobulin-like type 2 receptor alpha, which can be engaged by gD, gB or gH/gL[21-25]. The impact of glycans on the cellular receptors and the viral envelope proteins in modulating the distinct interactions are underexplored. Furthermore, while the importance of GAGs for HSV-1 initial attachment is well documented[26,27], little is known about the role of GSLs.

In this work we use HSV-1 as a model to systematically assess the importance of specific glycans in infection of a relevant human cellular system—the skin. We generate glycoengineered HaCaT keratinocytes by introducing targeted deletions in the cellular glycome, including select initiation, branching, and capping events, encompassing all the major classes of glycoconjugates (Fig. 1a). The use of live virus in its natural target cell type, representing the primary site of infection, allows us both to examine isolated viral life cycle events, such as binding, entry, propagation, and spread, and follow viral spread in 3D differentiated skin tissue models built with the glycoengineered cells. To identify specific glycan structures important in HSV-1 biology, we analyze isolated life cycle events using the library of glycoengineered cells. We find that each individual type of glycoconjugate affects at least one stage of HSV-1 life cycle and identify the critical glycan biosynthesis steps. We find chain-specific GAG sulfation, GSL sialylation and antenna-specific N-glycan branching important for attachment, O-glycan core synthesis important for propagation, and complex N-glycans important for cell-to-cell and tissue spread. We furthermore address roles of specific O-glycan acceptor amino acids on viral entry mediator gD and conserved fusogen gB. The work identifies the glycosylation steps that are essential for proper viral functions defining the scope of future mechanistic studies, but also suggests that local targeting of host glycosylation pathways by inhibitors or mimetics may be sufficient to inhibit redundant viral entry mechanisms.

## Results

### Generation of a glycogene knock out library in HaCaT keratinocytes

HaCaT is a human keratinocyte cell line capable of forming a stratified squamous epithelium, and thus allows evaluating the infection of the skin tropic HSV-1 in both cell and organotypic tissue culture. In order to address the role of specific glycan structures in the HSV-1 infectious cycle, we used precise gene editing to target glycosyltransferases involved in the early steps of core structure synthesis, and in major elongation and branching steps of the main glycosylation pathways, including N-linked glycosylation, mucin type O-linked glycosylation, as well as GSL and GAG synthesis (Fig. 1a, Supplementary Table 1).

For N-linked glycans we generated *MGAT1*, *MGAT4A*, *MGAT4B*, *MGAT5*, and *MGAT5 + 4B* knock outs (KO). MGAT1 adds the first N-acetylglucosamine to the C-2 of core α3-linked mannose, and lack of this enzyme results in elimination of all N-glycan maturation steps, yielding high-mannose type N-glycans as confirmed by MS-glycoprofiling (Fig. 1a, Supplementary Fig. 1). MGAT4A, MGAT4B and MGAT5 are responsible for N-glycan branching, where MGAT4A or MGAT4B initiate a β4-linked antenna on the α3-linked mannose, and MGAT5 performs β6-linked branching from the core α6-linked mannose. Lack of MGAT5 results in loss of tetra-antennary N-glycans, and loss of MGAT4 isoforms also strongly diminishes the content of tetra-

antennary N-glycans (Supplementary Fig. 1). In addition, KO of each of the three branching enzymes resulted in increased relative abundance of hybrid type N-glycans, whereas double KO of *MGAT5* and *MGAT4B* increased the relative abundance of biantennary glycans (Supplementary Fig. 1).

For mucin type O-linked glycans, we knocked out core 1 synthase (*C1GALT1*), its obligate chaperone COSMC (*C1GALT1C1*), core 2 synthase (*GCNT1*), as well as the major core 1-capping glycosyltransferase *ST3GAL1*. Loss of C1GALT1 or COSMC eliminates the β3-linked galactose (core 1 structure), results in truncation of O-linked glycans to the initiating α-GalNAc, and prevents assembly on secreted α-benzyl GalNAc precursor used in CORA O-glycoprofiling (Fig. 1a, Supplementary Fig. 2). GCNT1 is the predominant enzyme creating the branched core 2 structure by addition of β6-linked GlcNAc to the GalNAc. Loss of GCNT1 nearly abolished all the disialylated core 2 structures, though some structures matching the composition of monosialylated core 2 could still be detected. Such structures cannot be discriminated from isobaric core 1 structures, and a minor contribution from other GCNTs to core 2 synthesis cannot be excluded either. Finally, loss of ST3GAL1 significantly reduces the α3-linked sialic acid content and results in predominantly non-capped core 1 structures (Supplementary Fig. 2). We also targeted the synthesis of GSLs and GAGs. Through KO of *B4GALT5* or *ST3GAL5*, we generated cells with the truncated GSLs, glucosylceramide (GlcCer) and lactosylceramide (LacCer), respectively (Fig. 1a). Furthermore, we knocked out *B4GALT7*, which adds a β4-linked galactose to the initiating xylose in GAG biosynthesis, effectively truncating all classes of GAGs on membrane proteoglycans (Fig. 1a). The generated keratinocyte library represents a screening platform for roles of defined cell-surface presented glycan structures in HSV-1 biology in the context of natural infection.

### Propagation of HSV-1 in glycoengineered keratinocytes

To define the capacity of HSV-1 to complete the infectious cycle in glycoengineered keratinocytes, we infected confluent monolayers of the KO cell lines with HSV-1, and quantified HSV-1 DNA and infectious particles in the growth media at 17 h post infection (hpi) by qPCR and plaque titration, respectively. As a measure for viral replication fitness, we calculated the ratio of genome copies/infectious particles for each KO. When infecting cells with truncated O-glycans (*C1GALT1C1* KO) a decrease in viral titers was detected (Fig. 1b, e). In contrast, the same cells generated close to normal levels of viral DNA (Fig. 1c, f), suggesting decreased fitness of virions lacking elongated O-glycans (Fig. 1d, g). This feature was unique to complete truncation, and not seen when eliminating branching or sialylation of O-glycans. In cells lacking N-glycan maturation (*MGAT1* KO) we also found a lower number of infectious particles (Fig. 1b, e) with an apparent decreased fitness as indicated by an increase in the ratio of DNA/infectious particles (Fig. 1d, g). This apparent decrease in fitness was not detected in cells with loss of N-glycan branching, and in *MGAT4A* KO cells we even observed an overall increased viral output (Fig. 1b, c). When analysing cells with GSL synthesis defects, we found that lack of LacCer sialylation (*ST3GAL5* KO) accelerated virus production (Fig. 1b, c, e, f), but without any change in viral fitness (Fig. 1d, 1g). Finally, loss of cellular GAGs increased the production of viral particles (Fig. 1c). In conclusion, most of the tested glycogene disruptions permitted HSV-1 replication, and only disruption of N- or O-glycan maturation impaired viral fitness. We next evaluated the impact of defined glycan classes to distinct stages of the HSV-1 infectious cycle, including binding and entry, viral assembly and properties of progeny virus, and cell-to-cell spread.

### Binding and entry of HSV-1 to mutant cell lines

HSV-1 binds and enters human keratinocytes very rapidly, with around 30% of virions bound after 20 min on ice, and 80% after 2 h[28]. Most of the bound virions enter keratinocytes within 5 min after warming[28].

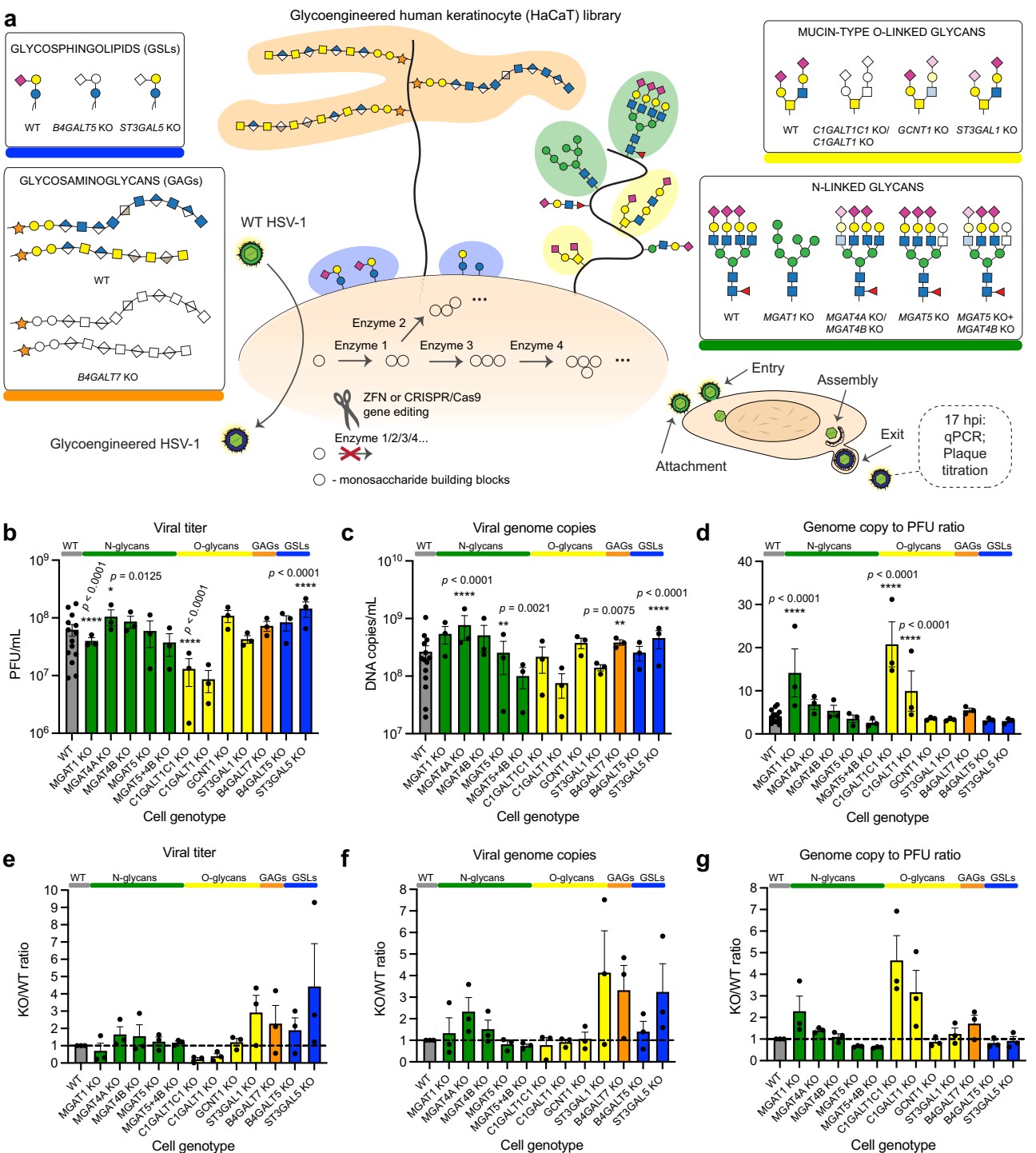

**Fig. 1 | Human keratinocyte library for probing host-pathogen interplay.**
**a** Common cell surface glycoconjugates are depicted, with investigated classes of glycans highlighted in boxes. Structures are drawn according to the symbol nomenclature for glycans (SNFG). The glycoengineered cell library was generated by knocking out genes involved in respective biosynthetic pathways using CRISPR/Cas9 and ZFN technologies. **b**–**d** The array of cell lines was infected with MOI 10 of HSV-1 Syn17 + , and viral titers (**b**) and viral DNA content (**c**) in the media were measured at 17 hpi. Particle to PFU ratio is also shown (**d**). Data points show average values ± SEM from 3 independent experiments for each knock out (KO) cell line including paired WT data from a total of 14 experiments. Two-way ANOVA followed by Dunnett's multiple comparison test was used to evaluate differences from WT (*$p < 0.05$, **$p < 0.01$, ***$p < 0.001$, ****$p < 0.0001$). **e**–**g** Data from (**b**–**d**) is shown as WT-normalized mean + SEM of 3 independent experiments for each KO cell line. Source data are provided as a Source Data file for all graphs.

Perturbations in each of the investigated glycosylation pathways modulated early virus-host interactions (Fig. 2a–e). Diminished core 2 O-glycan branching resulted in increased binding also reflected in subsequent entry experiments (Fig. 2b, c). Lack of complex N-glycans and reduced β4-antenna branching (*MGAT1* KO and *MGAT4B* KO) showed reduced binding, again also reflected in the entry experiments

(Fig. 2a–c). Interestingly, deletion of *MGAT4A*, another isoform catalyzing the β4-antenna synthesis on N-glycans, likely on another subset of proteins or sites in proteins[29,30], selectively affected viral entry (Fig. 2b, c, e). Cells displaying truncated glycolipids showed a reduction in binding to around 50% of that of WT (Fig. 2a, b). A similar effect was observed in both *B4GALT5* and *ST3GAL5* KOs, controlling the

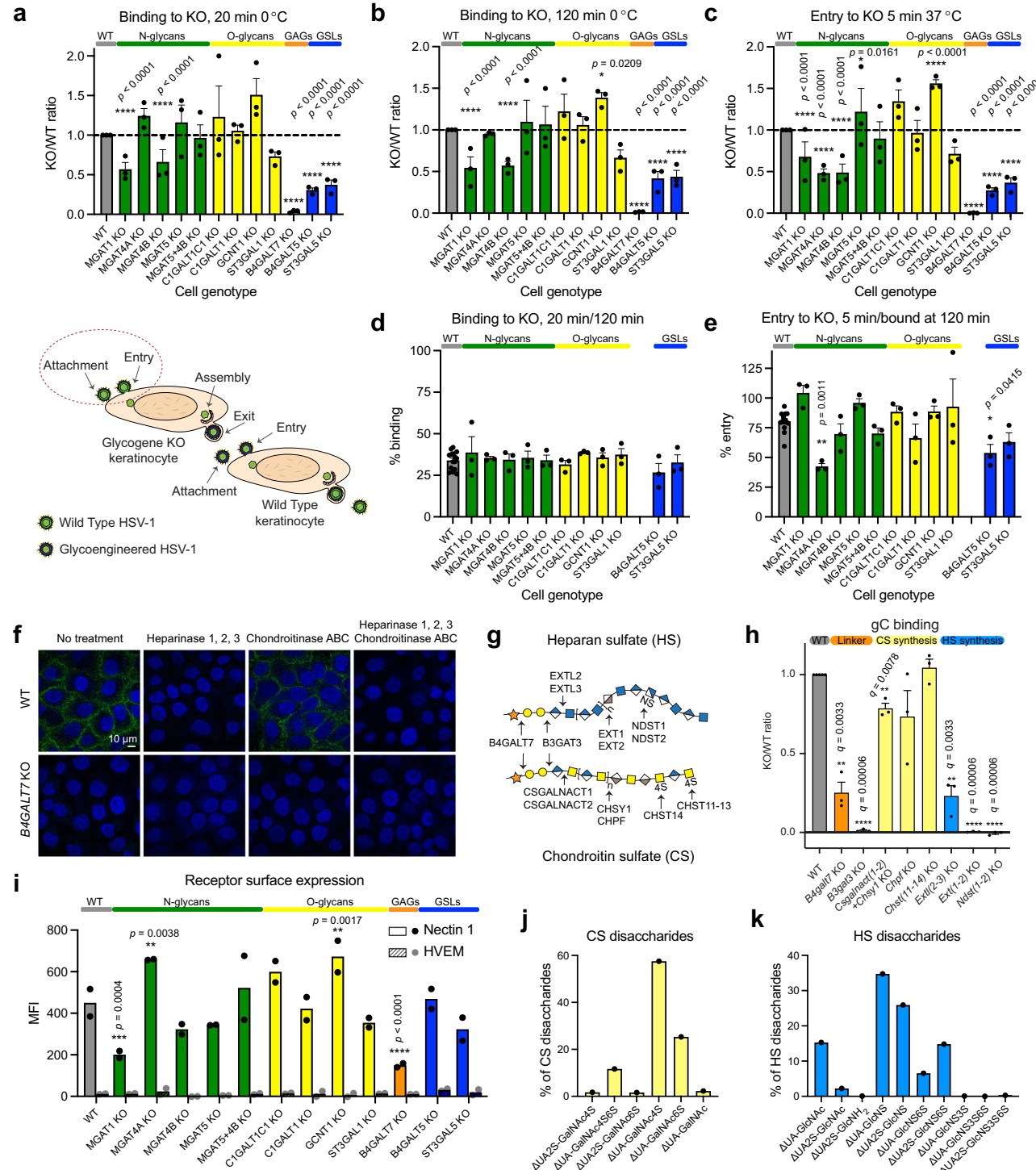

**Fig. 2 | Binding and entry of HSV-1 to mutant cell lines.** HSV-1 binding (20 min (**a**) or 120 min (**b**) on ice) and entry (5 min at 37 °C after 120 min on ice (**c**)) to KO cell lines. Data is shown as WT-normalized mean + SEM of 3 independent experiments for each KO cell line. Two-way ANOVA followed by Dunnett's multiple comparison test was used on raw data to evaluate differences from WT (*$p < 0.05$, **$p < 0.01$, ***$p < 0.001$, ****$p < 0.0001$). Proportion of virus bound at 20 min compared to 120 min (**d**) or proportion of virus entered at 5 min compared to virus bound at 120 min (**e**) is shown as mean + SEM of 3 independent experiments for each KO cell line from a total of 15 experiments. One-way ANOVA followed by Dunnett's multiple comparison test was used to evaluate differences from WT (*$p < 0.05$, **$p < 0.01$, ***$p < 0.001$, ****$p < 0.0001$). **f** HaCaT WT and *B4GALT7* KO cells were probed for HSV-1 gC binding. Enzymatic treatments were included to evaluate contributions of HS and CS GAG chains. Representative of 2 independent experiments. **g** A simplified overview of GAG synthesis. **h** HSV-1 gC binding to a panel of CHO KO cell lines. Data is shown as WT-normalized geometrical means of 3 independent experiments for each KO cell line, and the bar heights indicate mean + SEM. One sample *t* test was used to evaluate differences from 1. FDR at 5% was controlled by two-stage step-up method of Benjamini, Krieger and Yekutieli (*$q < 0.05$, **$q < 0.01$, ***$q < 0.001$, ****$q < 0.0001$). **i** Nectin 1 and HVEM surface expression. Data points show background subtracted median fluorescence intensity (MFI) from two independent experiments, and the bar heights indicate the mean. **j** Percentages of total quantified CS disaccharides in HaCaT WT (Supplementary Table 2). **k** Percentages of total quantified HS disaccharides in HaCaT WT (Supplementary Table 3). Source data are provided as a Source Data file for all graphs.

consecutive steps in the biosynthesis of the GSL GM3. In addition, an incremental reduction in entry was observed for *B4GALT5* KO cells, suggesting involvement of glycolipids in both viral binding and entry to host cells (Fig. 2c, e).

Then, we analysed cells impaired in GAG biosynthesis and found an almost complete loss of binding to cells presenting only the initiating xylose on proteoglycans (*B4GALT7* KO). Although we still lack a clear understanding of finer structural requirements of GAGs presented on their core proteins in the context of total cellular glycome, this fits well with the known importance of heparan sulfate (HS) in the initial attachment of HSV-1 (Fig. 2a, b)[26]. To further dissect the importance of GAG binding determinants we investigated the binding of recombinant HSV-1 gC to our HaCaT KO cells. The use of recombinant HSV-1 gC limited the interactions to a single viral protein known to bind to synthetic GAGs in vitro, similarly to intact HSV-1[27,31,32]. As expected, no binding was detected on *B4GALT7* KO cells (Fig. 2f), and to further confirm the selectivity for HS we treated HaCaT WT cells with heparinases 1, 2, and 3. Loss of HS completely abolished gC binding suggesting minimal interaction with chondroitin sulfate (CS) or dermatan sulfate (DS) presented on the cell surface (Fig. 2f). Minimal changes in cell staining for bound HSV-1 gC after chondroitinase ABC treatment further supported this interpretation. Next, we analyzed a library of glycoengineered CHO cells delineating the GAG biosynthesis pathways (Fig. 2g) and quantified gC binding by flow cytometry (Fig. 2h)[33]. This library included selective elimination of HS or CS (*Extl2* + *Extl3* KO and *Csgalnact1* KO + *Csgalnact2* KO + *Chsy1* KO, respectively), reduction in chain polymerization of HS or CS (*Ext1* KO + *Ext2* KO and *Chpf* KO, respectively), elimination of HS N-sulfation, also effectively diminishing follow-up O-sulfation (*Ndst1* KO + *Ndst2* KO), as well as elimination of 4-O sulfation of CS and DS units of CS chains (*Chst11* KO + *Chst12* KO + *Chst13* KO + *Chst14* KO). In addition, we used *B4galt7* KO and *B3gat3* KO cells to truncate all GAGs to the initiating xylose and a short linker trisaccharide, respectively (Fig. 2g). In agreement with the HaCaT cell staining data, manipulation of HS synthesis and chain length substantially decreased gC binding showing that the interaction was entirely dependent on HS sulfation and not compensated by the presence of CS (Fig. 2h). Accordingly, manipulation of CS synthesis only slightly decreased gC binding independent of the predominant 4-O sulfation (Fig. 2h). As expected, truncation to the linker also eliminated gC binding (Fig. 2h). To our surprise, some binding was retained upon complete GAG truncation, possibly representing unspecific binding due to gross changes in the glycocalyx. In conclusion, by using cell surface presented GAGs, we were able to identify sulfated HS as the major contributor to HSV-1 gC binding and show that CS sulfation is not necessary for interaction with CS, at least in the presence of HS. More generally, the binding and entry assays show that perturbations in the cellular glycome landscape have immediate effects to early virus-cell interactions, which can be further dissected as demonstrated for the interaction between gC and HS.

## The landscape of HSV-1 entry receptors

To follow up on our binding and entry data, we aimed to investigate the cellular landscape of HSV-1 entry receptors and other surface molecules that may have an impact on the early virus-cell interactions in the different knock out cells. We first quantified the surface expression levels of Nectin 1 and HVEM in WT HaCaT cells and found very low levels of the latter (Fig. 2i, Supplementary Fig. 3a, b). *MGAT1* KO and *B4GALT7* KO cells expressed significantly lower levels of Nectin 1 on the cell surface, whereas *MGAT4* KO and *GCNT1* KO expressed higher levels (Fig. 2i). These results correlate well with the virus binding data, and may help explain the altered proportion of virus bound to cells with alterations in N-glycosylation and O-glycosylation pathways. Importantly, the selective effect on entry to *MGAT4* KO was not correlated to availability of the receptor.

For *B4GALT7* KO, Nectin 1 presentation decreased by approximately 60%, but this does not explain the complete loss of HSV-1 binding, which is likely a combination of a decrease in GAG and protein receptors. While gC mediates early virus-GAG interactions, facilitating subsequent interactions between gD and its cognate protein entry receptors, 3-O-sulfated HS has also been identified as an independent entry receptor for gD[34,35]. In order to evaluate the potential contribution of 3-O-sulfated HS to HSV-1 entry in skin cells, we performed disaccharide analysis of HaCaT WT and *B4GALT7* KO cells, using our recently developed method, which allows detection of 3-O-sulfated HS[36] (Fig. 2 j, k, Supplementary Fig. 4, Supplementary Table 2 and 3). Except for hyaluronan, which is synthesized by a distinct family of enzymes, we did not detect any CS or HS disaccharides in *B4GALT7* KO cells (Supplementary Fig. 4). HaCaT WT cells expressed high levels of 4-O-sulfated or 6-O-sulfated CS, hyaluronan, as well as N-sulfated, N-/2-O-sulfated, N-/2-O/6-O-sulfated, and non-sulfated HS. We detected very low levels of 3-O-sulfated HS disaccharides, demonstrating that usage of these receptors for HSV-1 entry in human keratinocytes is limited. We therefore suggest that Nectin 1 is the most widely available HSV-1 entry receptor for gD in HaCaT keratinocytes.

The disaccharide expression profiles in skin cells provided additional insight into the gC binding data on the CHO cell library. Namely, N-sulfated GAG motifs required for gC binding to CHO cells were abundantly found on human keratinocytes, and likely play a significant role in vivo. On the contrary, 4-O-sulfated CS, abundantly found on skin cells, is unlikely to be a critical receptor for gC, as seen from CHO data.

We next looked into GSLs expressed in skin cells (Fig. 3). We saw comparable levels of Nectin 1 on the surface of WT, *B4GALT5* KO and *ST3GAL5* KO cells (Fig. 2i), and yet HSV-1 binding and entry to these cells was markedly decreased. We thus hypothesized that elongated GSLs may help deliver the viral entry receptors to membrane compartments accessible to incoming virus. We used antibodies and toxins recognizing various (glyco)lipid structures to illuminate their distribution in keratinocytes (Fig. 3a). Ceramide and glucosylceramide, representing initial steps of GSL synthesis, were predominantly located intracellularly in WT cells, while some ceramide accumulation could be seen in *B4GALT5* KO, devoid of elaborate GSLs (Fig. 3b). Interestingly, expression of more complex GSLs was heterogeneous, and different cells appeared committed to a specific GSL subtype. Specifically, we detected Gb3 structures, synthesized from lactosylceramide precursor, in both WT, and *ST3GAL5* KO cells with clear surface presentation, but not *B4GALT5* KO (Fig. 3b, e). In contrast, GM3, the product of *ST3GAL5*, was only detected in WT cells (Fig. 3b). GM3 partially co-localized with intracellular glucosylceramide-positive structures but were primarily expressed on the cell membrane (Fig. 3c). Importantly, GM3 was abundantly found on apical cell surfaces accessible to the extracellular environment (Fig. 3d). Gb3 and GM3 were expressed in mostly distinct subsets of cells, and a substantial proportion of skin cells remained unlabeled, presumably expressing more elaborate structures (Fig. 3e). In conclusion, we show heterogeneous yet regulated expression of different GSLs in distinct cells and within different cellular compartments, which may be relevant for interaction with extracellular virus.

## Viral assembly and properties of progeny virus

We next investigated late stages of viral replication in KO cells with changes in protein glycosylation capacity and altered viral propagation dynamics. We probed the expression of gD and gB that promote virion envelopment. In WT most of gD signal was confined to the cell surface, partially overlapping with E-cadherin (Fig. 4a), while gB primarily localized to the perinuclear compartment and secondary envelopment sites with some surface presentation, consistent with the literature (Fig. 4b)[37]. In contrast, *C1GALT1C1* KO, *C1GALT1* KO, and *MGAT1* KO cells exhibited a weaker and more dispersed gD immunostaining

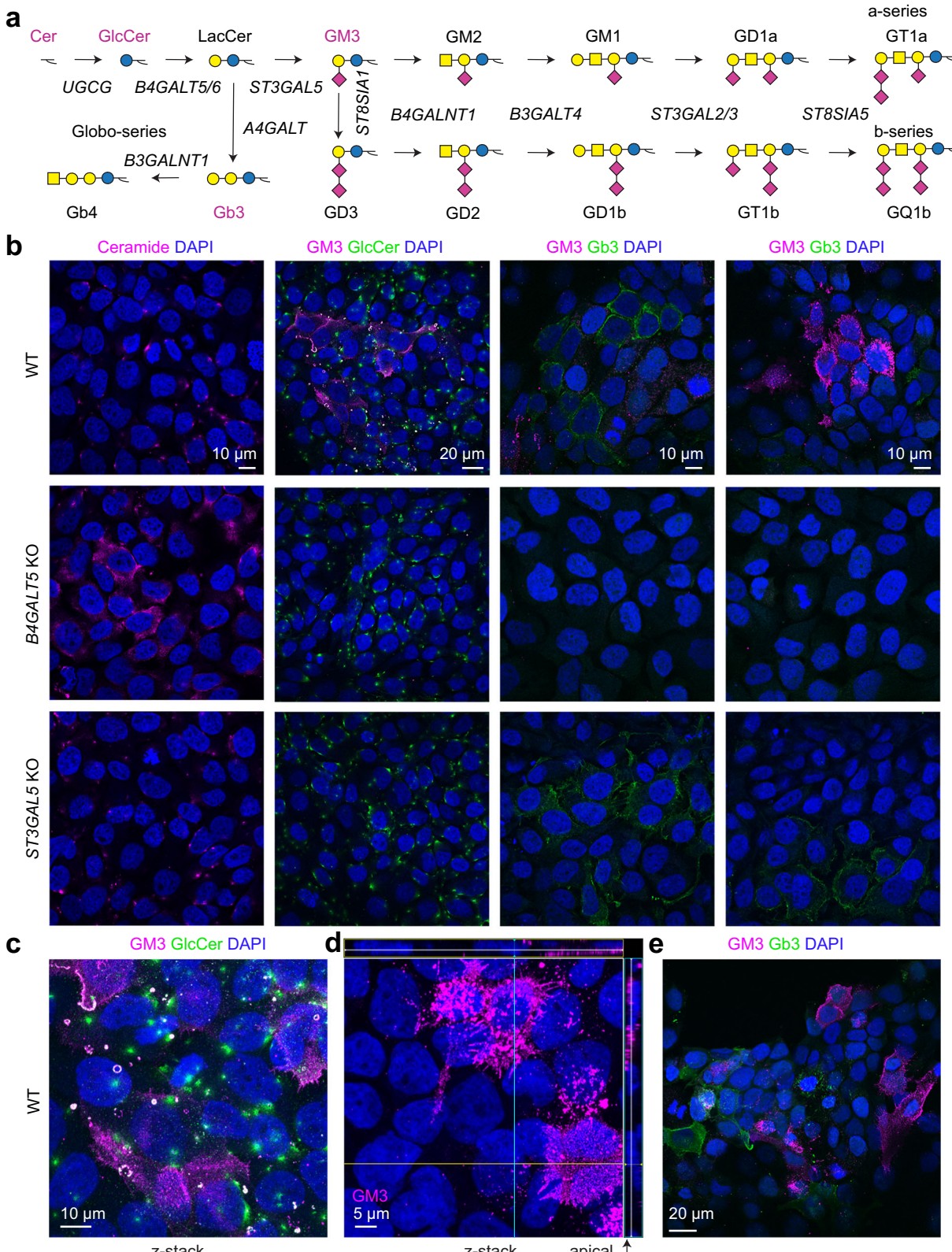

pattern with partial cytoplasmic accumulation suggesting issues with envelope glycoprotein trafficking (Fig. 4a). In addition, gB exhibited poorer surface and perinuclear localization and presented in large clusters within the cells (Fig. 4b). *ST3GAL1* KO cells, which did not exhibit defects in viral propagation dynamics, displayed similar gB staining as WT (Fig. 4b), while exhibiting stronger gD signal (Fig. 4a). Overall, the results suggest that lack of core 1 O-glycans or mature

N-glycans causes defects in viral particle formation due to suboptimal incorporation of viral proteins, which would fit with the observed diminished titers or loss of fitness. In addition, using an HSV-1 strain with GFP-labeled capsid protein VP26 allowed us to observe differences in the localization of viral capsids. The capsids were found in nuclear assembly compartments, outer nuclear rim, and transitioning through the cytosol in WT and *ST3GAL1* KO cells. We observed lower

**Fig. 3 | Heterogeneous expression of glycosphingolipids in HaCaT keratinocytes. a** The cartoon depicts a simplified human glycosphingolipid biosynthetic pathway. Glycolipid structures highlighted in magenta were probed by antibodies or fluorescently labeled toxins. **b**–**e** Cells grown on cover slips were fixed with 4% PFA and stained for different GSL structures. **b** Confocal micrographs show distribution of different GSLs in HaCaT WT, *B4GALT5* KO and *ST3GAL5* KO monolayers. Representative of two independent experiments, scale bars are indicated for each set of micrographs. **c** z-stack maximal intensity projection of HaCaT WT cells labeled with anti-GlcCer and anti-GM3 antibodies. Representative of 2 independent

experiments, scale bar is indicated. **d** HaCaT WT cells labeled with anti-GM3 antibody. An individual z-slice within a stack is shown, with orthogonal cross sections of the z-volume included, and indicate apical expression of GM3. Nuclei are labeled with DAPI (blue). Representative of 2 independent experiments, scale bar is indicated. **e** The confocal micrograph shows spatially distinct distribution of Gb3 and GM3 GSLs in HaCaT WT, probed by FITC-labeled Shiga toxin B (StxB-FITC), and anti-GM3 antibody, respectively. Representative of 2 independent experiments, scale bar is indicated.

numbers of capsid assembly sites in the nucleus and rare association with the outer nuclear rim in *C1GALT1C1* KO, *C1GALT1* KO and *MGAT1* KO cells, with the most pronounced effect in *C1GALT1C1* KO (Fig. 4a, b).

We further explored the viral replication dynamics in WT and *C1GALT1C1* KO cells by live imaging of GFP-labeled HSV-1. Features seen in thin optical sections (Fig. 4a, b) were also reflected in widefield images (Fig. 4c). In WT cells at 14 hpi, multiple capsid assembly sites could be seen in the nucleus and capsids were also associating with the nuclear envelope in most cells irrespective of the viral load (Fig. 4c). In *C1GALT1C1* KO cells less and smaller assembly sites could be seen, and capsids were less frequently associating with nuclear envelope. This association slightly improved later in infection (20 hpi), but the capsid production did not intensify, suggesting HSV-1 infection is generally less robust in *C1GALT1C1* KO (Fig. 4c).

To evaluate the contribution of viral glycans to fitness of progeny virus for early interactions with wild type host cells, we added equal numbers of infectious particles, produced in propagation experiments, to WT keratinocyte monolayers following the previously outlined strategy. No defects in binding or entry were found with virions lacking different glycan structures (Fig. 5a–e). In fact, virions lacking O-glycan elongation were capable of accelerated binding, despite low viral titers of HSV-1 produced in *C1GALT1C1* KO or *C1GALT1* KO (Fig. 5). This suggests the observed propagation defects are related to host and viral factors influencing the formation of infectious virions and not their efficiency in establishing a new infection.

## Role of site-specific O-glycosylation

The effect of O-glycosylation on HSV-1 glycoprotein localization, and prior knowledge of O-glycosite modifications compelled us to investigate specific O-glycosites. Eliminating site-specific O-glycosylation may have a more profound effect on protein function than truncation of the O-glycan structure[5,38,39]. Therefore, although O-glycan truncation had no deleterious effects on properties of infectious virions, it should not be excluded that individual O-glycosylation sites could play a functional role.

We have previously identified more than 70 O-glycosites on eight out of the 12 HSV-1 surface proteins, including the indispensable fusion machinery comprised of gB, gD, gH, and gL[15]. Based on available structural data and defined molecular mechanisms, we mutated five out of the identified 21 gB O-glycosites and three out of five gD O-glycosites most likely to affect fusion and receptor binding, respectively (Figs. 6a, 7a)[15]. We generated Ser/Thr to Ala substitutions alone or in combination to test cell-cell fusion efficiency using a split luciferase reporter assay as a proxy for viral entry (Supplementary Table 4, Supplementary Fig. 5). The assay quantifies fusion between two cell types, one (effector) lacking HSV-1 entry receptors and transfected with plasmids encoding the conserved fusion machinery, and the other (target) presenting HSV-1 entry receptors (Fig. 6b)[40]. Each cell type is also transfected with plasmids encoding half of a split luciferase reporter, which upon cell fusion can form a functional enzyme generating luminescence. In addition, we quantified gB and gD surface expression by CELISA[40]. We used CHO cells, refractory to HSV-1 entry, as effector, and HEK293, an HSV-1 permissive epithelial cell line, as target. We quantified low levels of Nectin 1 and HVEM on HEK293 cells (Supplementary

Fig. 3c), suggesting other types of receptors and co-receptors may also be involved.

For gB single site O-glycan mutants, we focused on the domain directly involved in fusion (I), where we identified three sites on anti-parallel beta strands (T169, T267, T268), as well as the arm domain (V) comprised of two alpha helices that undergo structural rearrangements upon fusion, where we found one O-glycosite on each (T690 and T703) (Fig. 6a)[15,41,42]. All mutations except for T169A and T268A affected gB cell surface expression; T267A and T703A showed moderate reduction, whereas T690A showed increased expression (Fig. 6c). T268A exhibited reduced fusion activity, as did T267A. Double or triple mutations in domain I severely decreased surface presentation and fusion activity (Fig. 6c, d, f, g). The activity of domain V single mutants did not correlate with changes in surface expression, where T690A exhibited very low fusion activity despite increased surface presentation (Fig. 6c, e, f, g). Interestingly, concomitant mutation of T703 (T690A T703A) could partially compensate for the strongly decreased activity of the T690A mutant (Fig. 6g).

Though gD does not directly execute fusion, it initiates entry by binding to several different host receptors and compromised interaction with gD would translate to reduced fusion efficiency. For gD, one O-glycan site on the N-terminal tail of the protein (S33 (8)), involved in interaction with both Nectin 1 and HVEM, and two O-glycan sites on an alpha helix undergoing structural changes upon interaction with HVEM (T255 (230) and S260 (235)), were mutated (Fig. 7a)[15,43–45]. All mutants maintained close to normal levels of cell surface expression of gD and fusion activity (Fig. 7b–d). To inspect possible contributions of gD mutations to interactions with distinct HSV-1 entry receptors, we utilized CHO cells overexpressing Nectin 1 or HVEM as target (Fig. 7e–h, Supplementary Fig. 3a, b). Here we saw a modest reduction in Nectin 1-initiated cell-cell fusion, when T255 and S260 were collectively mutated (Fig. 7e, f). A more pronounced reduction in cell-cell fusion efficiency was seen in HVEM-mediated entry upon introduction of these mutations (Fig. 7g, h).

In conclusion, we identified functionally relevant O-glycan acceptor amino acids on gB, directly executing fusion, but no effects were observed for the initial engager gD in the presence of multiple host entry receptors in HEK293 cells. However, compound mutations in gD affected isolated receptor-mediated entry.

## Spread of HSV-1 in mutant cell lines

Lastly, we investigated the roles of the specific classes of glycans in direct cell-to-cell spread mediated in part by gE/gI via cell contacts of 2D grown keratinocytes, and unrestricted spread in 3D skin culture, facilitated by tissue destruction and release of free virions (Fig. 8a).

We first performed plaque assays with 2D grown cells, where dissociation of progeny virions is impeded by the dense overlay media, making direct cell-to-cell spread as the predominant mode of spread. Perturbations in core 1 O-glycan biosynthesis resulted in increased plaque size, most notably in *C1GALT1* KO and *ST3GAL1* KO cells (Fig. 8b). Upon plaque immunostaining, WT cells and KO cells exhibiting increased plaque size showed strong gE expression on the cell surface (Fig. 8c). In cells lacking N-linked glycan maturation (*MGAT1* KO) and those lacking MGAT4B (*MGAT4B* KO; *MGAT5 + MGAT4B* KO), resulting in reduced β4-antenna branching, we found a markedly

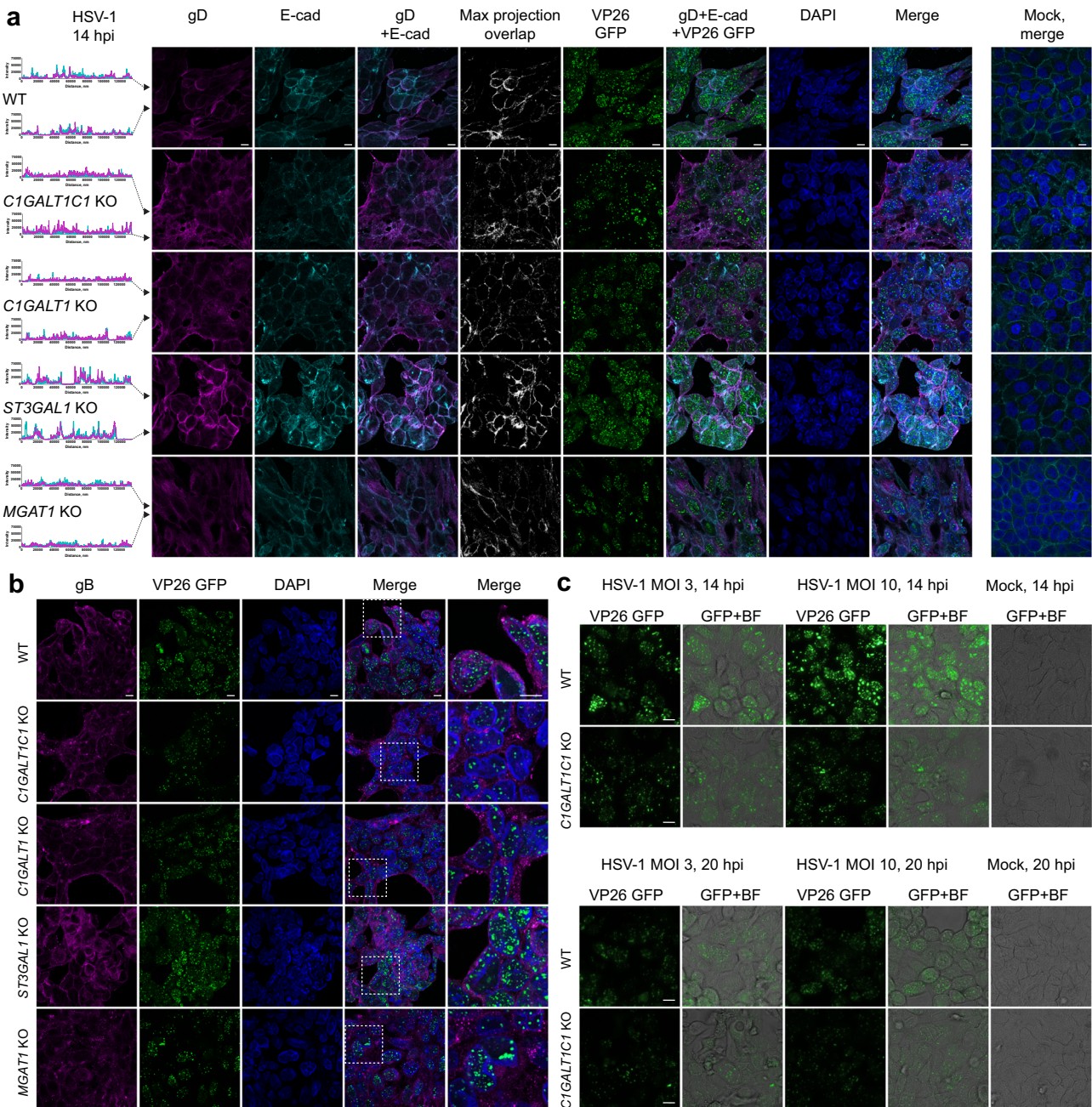

**Fig. 4 | Viral assembly in glycoengineered cells. a** HaCaT cells grown on cover slips were infected with MOI10 of HSV-1 K26-GFP and fixed and permeabilized at 14 hpi followed by co-staining for HSV-1 gD (magenta) and E-cadherin (cyan). Histograms on the left indicate intensities of gD and E-cadherin signals across the confocal images (marked with black arrowheads). Pixel overlap from the two channels is shown in white. GFP labeled capsid proteins (VP26) are seen in green. Nuclei were stained with DAPI (blue). Stainings of mock-infected cells are included. Scale bar: 10 μm. Images are representative of two independent experiments. **b** HaCaT cells grown on cover slips were infected with MOI10 of HSV-1 K26-GFP and

fixed and permeabilized at 14 hpi followed by staining for HSV-1 gB (magenta). GFP labeled capsid proteins (VP26) are seen in green. Nuclei were stained with DAPI (blue). Scale bar: 10 μm. Magnified regions of merged images are indicated with dashed white boxes. Images are representative of 2 independent experiments. **c** HaCaT WT and *C1GALT1C1* KO cells were infected with MOI3 or MOI10 of HSV-1 K26-GFP and viral capsids imaged by live microscopy at 14 and 20 hpi. Fluorescent images overlaid with bright field images are also shown. Scale bar: 10 μm. Images are representative of two independent experiments.

reduced cell-to-cell spread (Fig. 8b). *MGAT1* KO cells showed less pronounced and more punctate gE expression, which may be linked to N-glycosylation sites on gE and help explain the limited spread capacity. Surprisingly, accelerated spread was observed in *MGAT4A* KO cells, which also contributes to β4-antenna branching, and a similar tendency was observed for *MGAT5* KO, devoid in β6-linked antenna branching.

To assess viral spread in tissue, we infected fully developed 3D epidermises built with the glycoengineered cells (Fig. 9a). Different

spread characteristics were observed, when viral spread was not limited to cell-to-cell contacts mediated by gE/gI complex. In wild type HaCaT skin equivalents trans epidermal lesions were observed at 36 hpi (Fig. 9b, c). To avoid selection bias, we generated ten subsequent tissue sections separated by 30 microns and scanned whole sections, which allowed to visualize and compare the extent of the viral lesions (Fig. 9a, b, Supplementary Fig. 6). We identified lesions spanning several sections and measured the cross-section areas corresponding to the central regions of those lesions (Fig. 9d). Large lesions were seen in

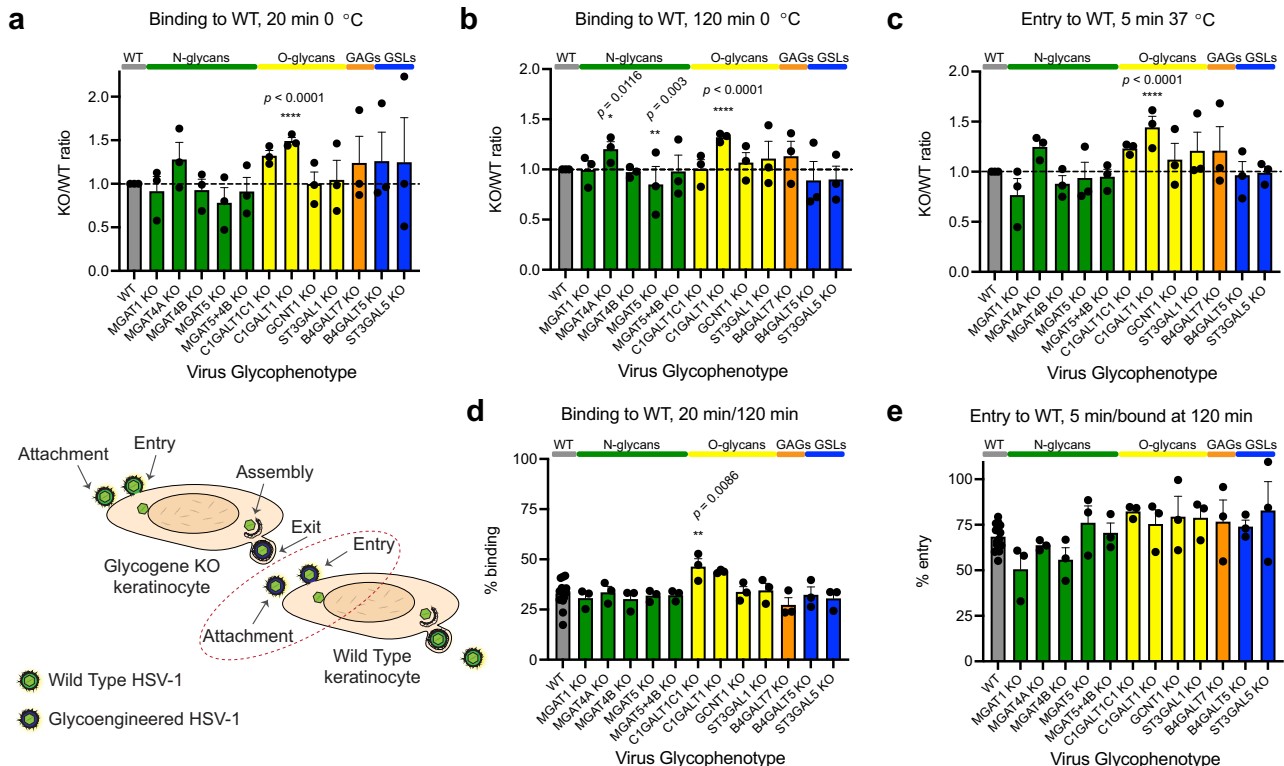

**Fig. 5 | Properties of glycoengineered virions.** Binding (20 min (**a**) or 120 min (**b**) on ice) and entry (5 min at 37 °C after 120 min on ice (**c**)) of HSV-1 produced in different KO cell lines to HaCaT WT. Data is shown as WT-normalized mean + SEM of three independent experiments for each glycoengineered virus species. Two-way ANOVA followed by Dunnett's multiple comparison test was used on raw data to evaluate differences from WT (*$p < 0.05$, **$p < 0.01$, ***$p < 0.001$, ****$p < 0.0001$). Proportion of virus bound at 20 min compared to 120 min (**d**) or proportion of virus entered at 5 min compared to virus bound at 120 min (**e**) is shown as mean + SEM of three independent experiments for each glycoengineered virus species from a total of 14 experiments. One-way ANOVA followed by Dunnett's multiple comparison test was used to evaluate differences from WT (*$p < 0.05$, **$p < 0.01$, ***$p < 0.001$, ****$p < 0.0001$). Source data are provided as a Source Data file for all graphs.

*MGAT1* KO tissues, contrasting the small plaques observed in 2D (Fig. 9c, d). Most N-glycan branching KO tissues, especially *MGAT4A* KO and *MGAT5 + 4B* KO, permitted only limited spread in the top layers of the epidermis. *MGAT4B* KO allowed formation of bigger lesions, but the tissue penetrance was limited, which was also the case for tissues with reduced core 1 sialylation (*ST3GAL1* KO) (Fig. 9c, d). No significant spread defects were noted for tissues with disruptions in GSL and GAG synthesis.

## Discussion

We utilized a genetically engineered keratinocyte library to systematically investigate the contributions of different classes of glycan structures to multiple stages of HSV-1 life cycle in the natural context of infection and investigated the properties of glycoengineered virions. This approach allowed the discovery that each class of glycans modulated select stages of the viral life cycle, and we narrowed down the critical biosynthetic steps mediating these effects. We found chain-specific GAG sulfation, GSL sialylation and antenna-specific N-glycan branching important for attachment, O-glycan core synthesis important for propagation, and complex N-glycans important for cell-to-cell and tissue spread. In addition, reducing the complexity of the O-glycome potentiated glycoengineered virion attachment, and cell-to-cell spread. Since distinct glycan classes could independently fine-tune early virus-host interactions, we suggest that HSV-1 through its evolution developed elaborated mechanisms to exploit the host glycosylation machinery[46].

We obtained the most unequivocal effects on HSV-1 infection upon GAG- and GSL-associated perturbations in the glycocalyx. Most previous studies investigating the roles of GAGs and GSLs are based on isolated systems involving synthetic (glyco)conjugates and isolated or overexpressed viral proteins, often in non-permissive cells. While HSV-1 has been shown to interact with both terminal HS and CS glycan chains in solution or on solid support[27, 31,32], data on HSV-1 interaction with GAGs attached to their intrinsic core proteins and presented in their native context of the glycocalyx is only beginning to emerge[47]. We here demonstrated that binding of viral particles to human keratinocyte monolayers lacking GAG moieties on proteoglycans on ice was almost completely inhibited, emphasizing the role of GAGs in very early interactions. Despite of this, HSV-1 propagated well in *B4GALT7* KO cells. This agrees with HSV-1 being able to infect mouse cells almost devoid of GAGs (sog9), especially at higher viral loads[48], and with increased release of virions from cells upon overexpression of heparanase[49]. This underscores the contextualized functions of glycans as modulators of molecular interactions, which only become obligate under specific stress conditions[6]. Our findings complement the well documented role of GAGs in herpesvirus biology[26,27,31,50–52]. It is proposed that GAGs allow HSV particles to "surf" on the cell surface until encountering cognate receptors for gD and initiating entry. Lack of such interactions has been demonstrated to prevent free movement of the viral particles on membrane equivalents or the surface of cells, and distinct roles of HS and CS in promoting lateral diffusion or confinement has recently been suggested[47,51]. By using a genetic approach, we determined that sulfated HS is the major contributor of HSV-1 gC binding to CHO cells, with CS not able to sustain the interaction in the absence of HS. By selective desulfation of purified HS, it has previously been shown that 2-O, 6-O, and N-sulfation, and at least a fragment of 12 monosaccharides is needed for optimal gC-HS interaction[32]. gC was still able to bind de-N-sulfated HS partially[32], confirming that N-sulfation, genetic depletion of which completely abolished gC binding in our study, is also important for creation of sulfate-rich GAG patches by

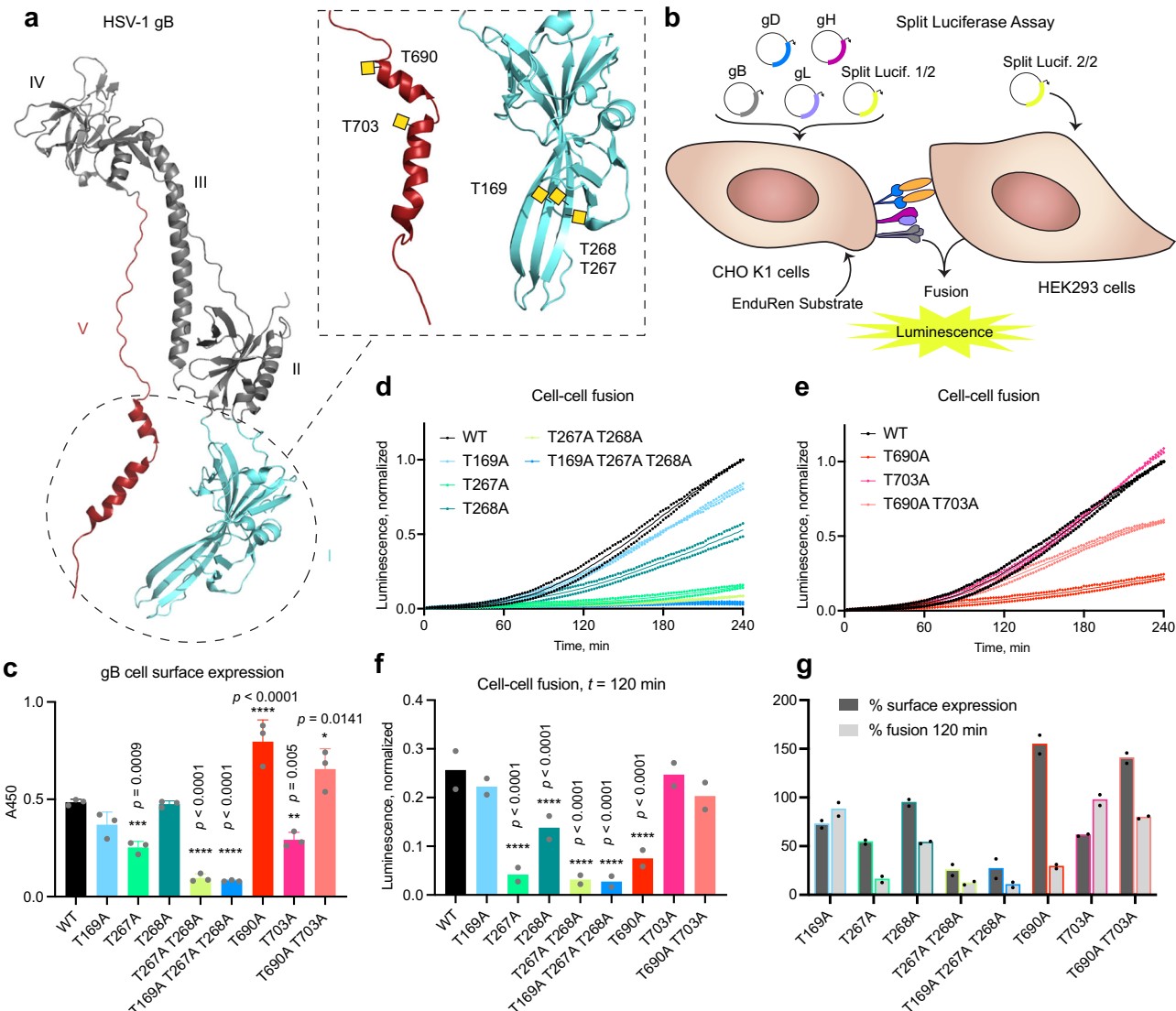

**Fig. 6 | Single O-glycosites affect gB-mediated cell-cell fusion efficiency. a** HSV-1 gB structure (PDB: 2GUM) with select mutated O-glycan acceptor sites indicated within the dashed box. Respective previously identified O-glycans were drawn manually as yellow squares. Domains are numbered in roman numericals according to Heldwein et al., Science 2006. **b** The cartoon illustrates the principle of split luciferase assay. **c** Cell surface expression of gB O-glycosite Thr to Ala mutants evaluated by CELISA using mouse anti-gB antibodies. Data is shown as mean absorbance at 450 nm + SD of three technical replicates and is representative of three independent experiments. One-way ANOVA followed by Dunnett's multiple comparison test was used to evaluate differences from WT (*$p < 0.05$, **$p < 0.01$, ***$p < 0.001$, ****$p < 0.0001$). **d, e** Cell-cell fusion activity over 240 min using gB O-glycosite Thr to Ala mutants. Data from two independent experiments is shown, where mean normalized luminescence of three technical replicates at each time point is indicated by a dot. Mean values of the two independent experiments are shown as thin lines. Data is normalized to maximum luminescence reading at final time point using WT gB for each experiment. **d** Data related to gB domain I mutations. **e** Data related to gB domain V mutations. **f** Cell-cell fusion activity of gB mutants at $t = 120$ min. Data is shown as mean normalized luminescence from two independent experiments. Two-way ANOVA followed by Dunnett's multiple comparison test was used to evaluate differences from WT (*$p < 0.05$, **$p < 0.01$, ***$p < 0.001$, ****$p < 0.0001$). **g** Average percentages of cell surface expression and fusion efficiency at $t = 120$ min from two independent experiments are shown in side-by-side columns. Source data are provided as a Source Data file for all graphs.

follow-up 2-O and 6-O sulfation. Our GAG disaccharide analysis on HaCaT cells identified abundant levels of 2-O, 6-O, and N-sulfated HS disaccharides, confirming these ligands are available on the natural target cell type.

After attachment, HSV-1 enters keratinocytes either via direct membrane fusion or endocytosis, and both of these routes are thought to be cholesterol-rich lipid raft dependent and involving interactions between viral and cellular proteins[25,28,53–55]. Mechanistic insight into the interaction of sphingomyelin and cholesterol with HSV-1 gH leading to membrane fusion is available[56]. In contrast, knowledge on the role of GSL, another major component of lipid rafts, in HSV-1 biology is scarce. Here, we show that elongation of host GSLs beyond glucosylceramide is important for efficient viral binding and entry. It is suggested that

αVβ3-integrin reroutes Nectin 1 to lipid rafts thus promoting cholesterol-mediated entry[57]. It is therefore plausible that altering the composition of lipid rafts by changing the GSL content may interfere with proper Nectin 1 trafficking as well as membrane fusion dynamics via gH. Our data suggest that Nectin 1, and not HVEM or 3-O-sulphated HS, is the most abundant gD entry receptor on HaCaT keratinocytes. Furthermore, we found highly heterogeneous distribution of different types of GSLs in cell monolayers and within subcellular compartments warranting future investigations into the likely complex regulatory mechanisms.

We identified diverse effects on the infectious cycle attributable to select maturation or branching steps of protein-bound N- and O-linked glycans. Investigating the importance of complex N-linked

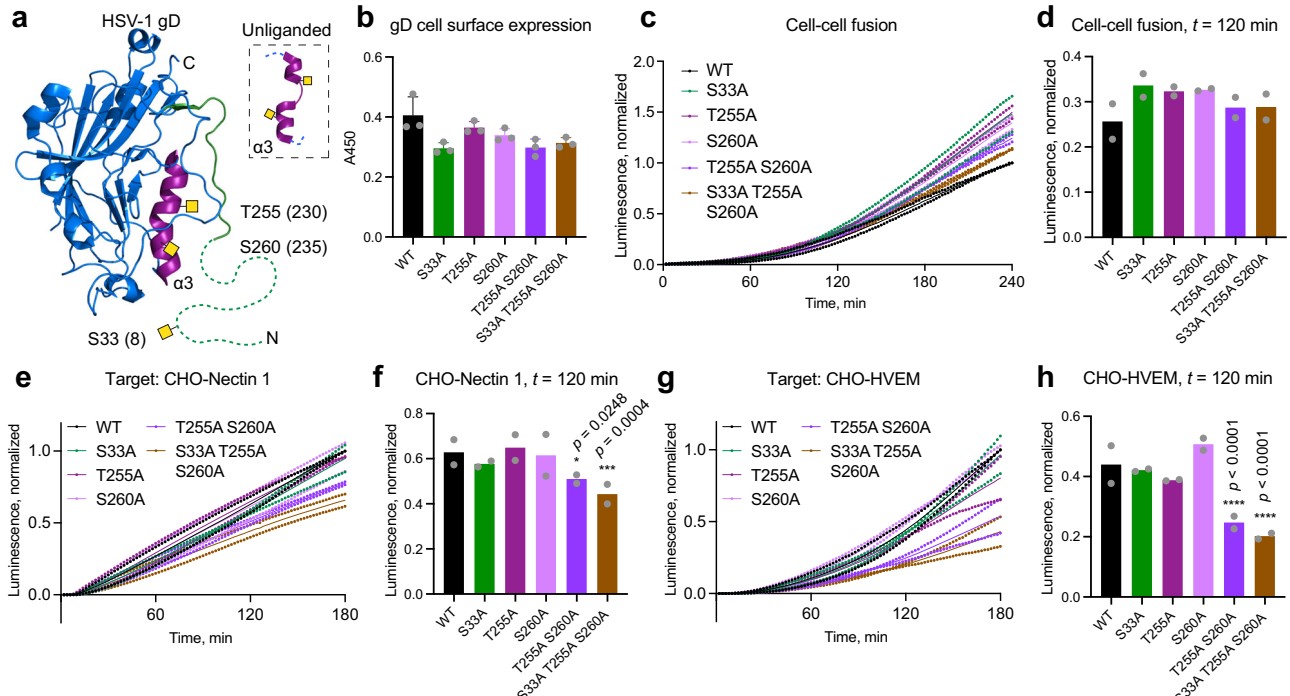

**Fig. 7 | Single O-glycosites contribute to gD-initiated cell-cell fusion via distinct receptors. a** HSV-1 gD structure (PDB: 2C36) with select mutated O-glycan acceptor sites indicated. Positions after removal of signal peptide, often encountered in the literature, are indicated in brackets. Respective previously identified O-glycans were drawn manually as yellow squares. N-terminal region omitted in the crystal structure is drawn as a dashed line. **b** Cell surface expression of gD O-glycosite mutants evaluated by CELISA. Data is shown as mean absorbance at 450 nm +SD of three technical replicates and is representative of three independent experiments. One-way ANOVA followed by Dunnett's multiple comparison test was used to evaluate differences from WT. **c** Cell-cell fusion activity over 240 min using gD O-glycosite mutants. Data from two independent experiments is shown, where mean normalized luminescence of three technical replicates at each time point is indicated by a dot. Mean values of the two independent experiments are shown as thin lines. Data is normalized to maximum luminescence reading at final time point using WT gD for each experiment. **d** Cell-cell fusion activity of gD mutants at $t = 120$ min. Data is shown as mean normalized luminescence from two independent experiments. Two-way ANOVA followed by Dunnett's multiple comparison test was used to evaluate differences from WT. CHO cells stably expressing Nectin 1 (**e**, **f**) or HVEM (**g**, **h**) were used as target cells to evaluate cell-cell fusion activity using gD O-glycosite mutants. Cell-cell fusion activity over 180 min using CHO-Nectin 1 (**e**) or CHO-HVEM (**g**) as target. Parental CHO cell line without entry receptors was use for background subtraction. Data is presented as in (**c**). Cell-cell fusion activity of gD mutants at $t = 120$ min using CHO-Nectin 1 (**f**) or CHO-HVEM (**h**) as target. Data is shown as mean normalized luminescence from two independent experiments. Two-way ANOVA followed by Dunnett's multiple comparison test was used to evaluate differences from WT (*$p < 0.05$, **$p < 0.01$, ***$p < 0.001$, ****$p < 0.0001$). Source data are provided as a Source Data file for all graphs.

glycans, we saw significantly smaller plaque size in *MGAT1* KO cells, indicating diminished cell-to-cell spread, while viral titers were only marginally reduced. Both observations are in accordance with previous investigations in a ricin-resistant baby hamster kidney cell line RicR14, deficient in Mgat1 activity[58]. In contrast to the previous studies, we did, however, observe an increase in the ratio of viral DNA to infectious particles, and an altered localization of envelope glycoproteins gB and gD, suggestive of less productive assembly in cells lacking complex N-glycans. Furthermore, we detected lower adsorption of HSV-1 to *MGAT1* KO cells. In another ricin resistant cell line, Ric21, presenting with N-linked glycans of reduced complexity and sialylation, HSV-1 infection also showed smaller plaque size and retarded particle production[59], suggesting the effects we see in *MGAT1* KO cells may be related to lack of sialylation. Accordingly, treatment of HSV-1 viral particles with α2,6 specific neuraminidase has previously been shown to dramatically reduce the infectivity of viral particles[60]. However, we did not observe diminished binding of infectious particles bearing immature N-glycans lacking sialylation despite N-linked glycans being the major acceptors of α2,6-linked sialic acids in keratinocytes.

The use of cell lines defective in N-glycan branching allowed us to differentiate between contribution of β4-linked and β6-linked antennae and compare it to complete loss of N-glycan maturation. While we did not see any importance of β6-linked antennae, generated by MGAT5, we found decreased binding of wild type HSV-1 to and

diminished cell-to-cell spread in *MGAT4B* KO cells, which is the main contributor to β4-linked antenna synthesis. Interestingly, MGAT4A, another isoform capable of generating β4-linked antennae, selectively reduced entry to host cells and dramatically increased HSV-1 spread in mutant cell monolayers. MGAT4A and MGAT4B are differentially expressed with MGAT4B being most abundant[61] and differential substrate protein preferences have recently been shown in vitro possibly regulated via the C-terminal lectin domain[29,30]. The distinct effects of the two enzymes are also demonstrated in *MGAT4A* KO mice presenting with hyperglycemia due to selective N-glycan branching of the GLUT2 transporter by MGAT4A, despite the co-expression of MGAT4B in the pancreas[61,62]. Moreover, glycosyltransferases have been described to form complexes with each other and with nucleotide sugar transporters in the Golgi membrane, and it has been shown that MGAT1 interacts with MGAT4B, but not MGAT5[63]. It is therefore possible, that such complex formation or lack thereof is affecting secretory pathway dynamics and protein substrate selectivity. This may help explain the comparable effects on virus-cell binding and cell-cell spread for *MGAT1* KO and *MGAT4B* KO suggesting importance of select N-glycosylation by MGAT4B of specific cellular proteins, and the distinct phenotypes caused by KO of *MGAT4A*. We saw diminished Nectin 1 surface expression in *MGAT1* KO cells, which also helps explain the reduced binding.

We previously demonstrated the importance of mucin type O-glycan elongation for HSV-1 propagation and early immune

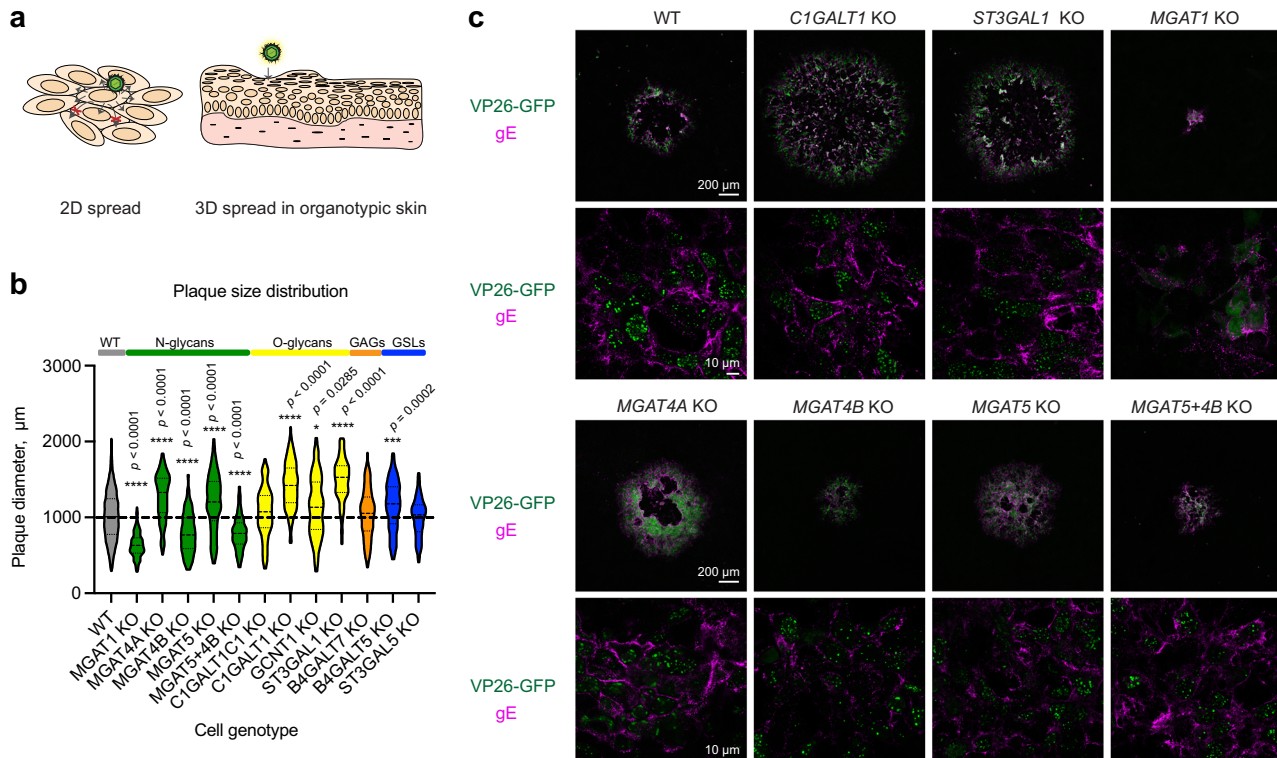

**Fig. 8 | HSV-1 cell-to-cell spread in mutant cell monolayers. a** The cartoon illustrates different modes of HSV-1 cell-to-cell spread in 2D glycoengineered HaCaT cell monolayers in the context of a plaque assay, and spread in 3D organotypic skin models. **b** Plaque diameter on cell monolayers infected with HSV-1 Syn17+ at 48 hpi. Data is presented as violin diagrams that include measurements from 3 independent experiments for each KO cell line, with 50 plaques measured for each experiment. Paired WT data includes measurements from 15 independent experiments. The dashed lines within the plots indicate median diameter, whereas the dotted lines indicate the interquartile range. One-way ANOVA followed by Games-Howell's multiple comparison test was used to evaluate differences from WT (*$p < 0.05$, **$p < 0.01$, ***$p < 0.001$, ****$p < 0.0001$). Source data are provided as a Source Data file. **c** Cell monolayers grown on cover slips were infected with 200 PFU (MOI < 0.0005) of HSV-1 K26-GFP and overlaid with semi-solid media for 48 h followed by fixation and staining for HSV-1 gE (magenta). GFP labeled capsid proteins (VP26) are seen in green. Confocal images at two different magnifications were taken to illustrate overviews of plaques (4 combined tiles at 10x, upper panels) as well as gE expression at higher resolution (63x, lower panels). Scale bars for the different magnifications are indicated.

sensing[15,64]. Here, O-glycan biosynthesis was dissected to include branching and capping steps. We discovered that the decreased viral titers and fitness were associated with complete truncation of O-glycans to α-GalNAc by KO of *C1GALT1* or its chaperone *C1GALT1C1*, and not with loss of sialylation of core 1 structures by ST3GAL1 or reduced core 2 branching by GCNT1. In fact, lack of core 2 structures on host cells resulted in accelerated binding and thus entry of WT virus to cells, correlating with increased Nectin 1 surface presentation. Since core 2 structures comprise a significant portion of the cellular O-glycome, it is also plausible that the reduced complexity on the cell surface would allow better accessibility. Loss of O-glycan elongation affected glycoprotein localization, with lower gD surface presentation, and altered gB localization, possibly influencing their incorporation into viral particles in turn affecting the proportion of particles capable of productive infection. In addition, gB is needed for de-envelopment at the outer nuclear membrane[65], and perturbed localization due to loss of O-glycans could contribute to lower release rate of capsids to the cytosol. In contrast, loss of core 1 elongation or sialylation allowed for accelerated cell-to-cell spread mediated by HSV-1 gE and gI, where competent surface expression of the former protein suggests protein-selective effects of O-glycans for proper localization.

We have previously shown that HSV-1 and other herpesviruses are heavily O-glycosylated[15,66]. To investigate the function of O-glycans at specific sites we used a cell-based assay probing the function of viral entry machinery consisting of gB, gD, gH and gL. We focused on specific O-glycan acceptor amino acids on gB and gD. In the fusion effector gB we selected three O-glycosites (T169, T267, T268)

decorating the central region of the beta sheet terminating in fusion loops (domain I). In addition, we examined two O-glycosites (T690, T703) present on the internal alpha helices important for reinforcing trimer interactions and undergoing extensive reorientation upon fusion (domain V)[15,41,42]. We hypothesized that the glycans, due to their critical positioning, could modulate the conformational stability and rearrangements of their respective domains. When eliminating specific O-glycan acceptor sites within domains directly or indirectly involved in fusion (I and V, respectively) we saw a clear effect on gB function and cell surface expression. Some of the mutations diminished gB-mediated cell-cell fusion (T267A, T268A, T690A). One of the positions (T690) has previously been identified to reduce gB function by linker insertion mutagenesis, where reduced cell surface expression was also seen[67]. This is in contrast to our observed increased surface presentation of T690A mutant. Interestingly, an additional glycosite mutation in domain V (T690A T703A) could partially rescue the strongly diminished activity of the T690A mutant. Hyperfusogenic mutations in the gB ectodomain are rare, while such mutations are often found in the cytoplasmic tail[68,69]. Hyperfusogenic mutations have been shown to affect the stability of the trimer in the pre-fusion conformation and the energy barrier threshold for executing fusion[70]. It will be important to validate the effect is truly glycan dependent and not influenced by the change in amino acid itself, and to define O-glycan occupancy on HSV-1 virions. Nevertheless, the work identifies several hotspots in gB structure that could be targeted for disrupting protein function. In contrast, infectious virions bearing truncated O-glycans did not exhibit any defects in binding or entry; in fact, we

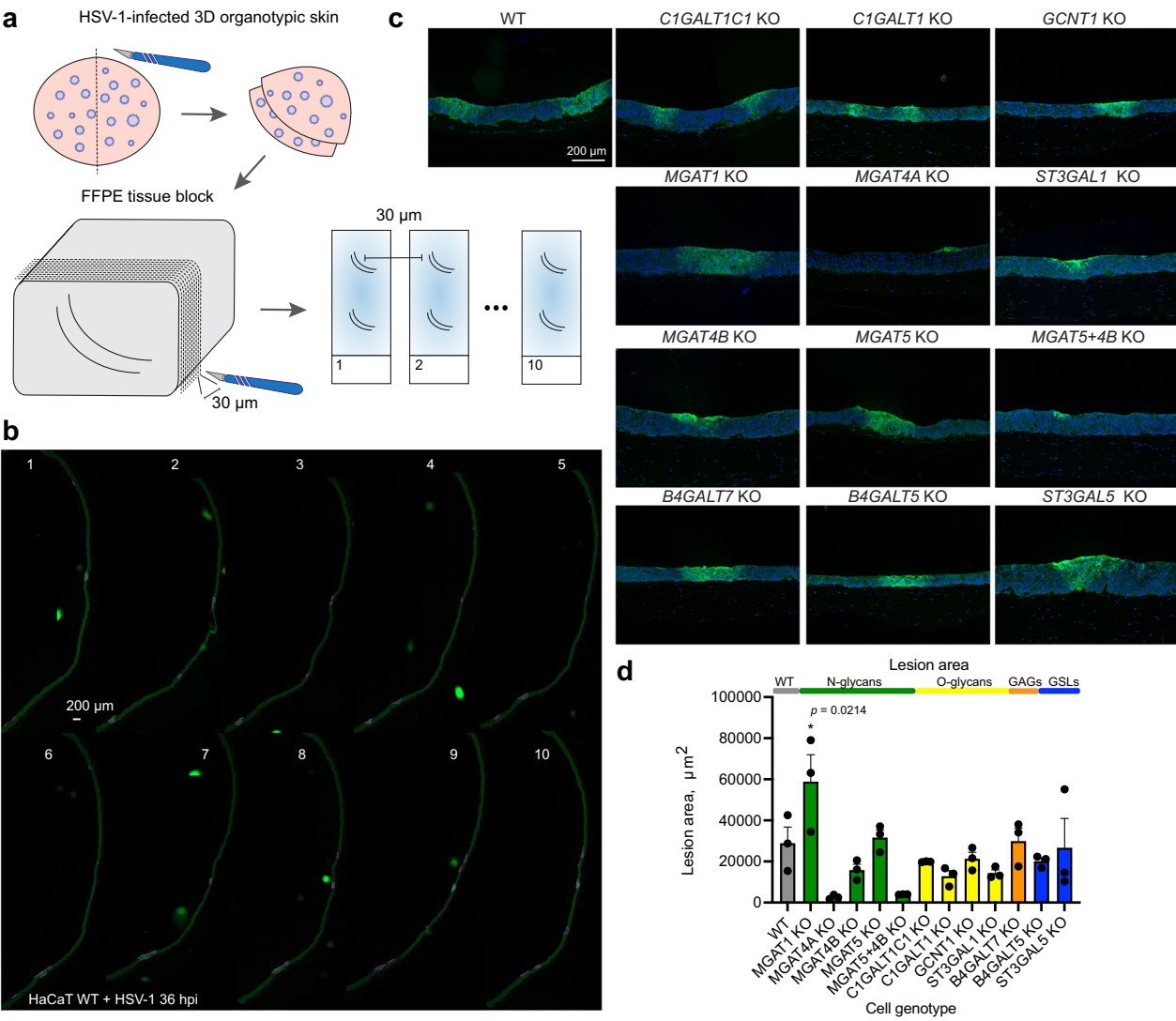

**Fig. 9 | HSV-1 cell-to-cell spread in mutant cultured tissues. a–d** Fully differentiated 3D skin models built with glycoengineered cells were infected with HSV-1 Syn17+ for 36 h followed by fixation in formalin and embedding in paraffin. **a** The cartoon illustrates the procedure for evaluating HSV-1 spread in organotypic skin tissues. FFPE tissues were sectioned every 30 μm for 10 consecutive slices containing two sections each, spanning from the center of the tissue outwards in two directions. **b** Consecutive sections were stained with a polyclonal FITC-labeled anti-HSV-1 antibody to visualize virus lesions and whole sections imaged with a microscope slide scanner. An example section series is shown with HSV-1 lesions outlined in purple. **c** Representative lesions were selected from the scanned section series for each KO tissue. Nuclei were labeled with DAPI. **d** Three lesions spanning several sections were identified for each KO tissue and lesion area measured at the centermost section. Data is shown as mean + SEM with individual measurement values indicated as black dots. One-way ANOVA followed by Dunnett's multiple comparison test was used to evaluate differences from WT (*$p < 0.05$, **$p < 0.01$, ***$p < 0.001$, ****$p < 0.0001$). Source data are provided as a Source Data file.

observed accelerated binding, suggesting that even very short O-glycans are sufficient for maintaining functions of properly localized proteins. In contrast, eliminating site-specific glycosylation on gD did not diminish cell-cell fusion. Since HEK293 cells likely express multiple receptors for gD, alternative receptor usage likely would rescue inefficient binding to one of the receptors. We additionally utilized CHO cells overexpressing Nectin 1 or HVEM as target, and for the latter found substantially lower cell-cell fusion activity when gD carried multiple mutations in the alpha helix important for interaction with HVEM.

In further attempts to examine contextualized functions of glycans in a biologically relevant system, we utilized the ability of keratinocytes to form stratified epithelia to investigate viral spread in glycoengineered tissue. Strikingly, we saw very different effects of glycans in 3D tissue spread compared to overlaid monolayers. This suggests the effects we see in 2D are directly related to importance of glycans on gE/gI complex or their receptors, whereas 3D spread involves both cell-to-cell and virus-to-cell transmission, and is also influenced by changes in tissue architecture induced by glycogene KOs. It should be emphasized that both the cells and the spreading virus are glycoengineered in such setups, and pinpointing the viral or the host protein responsible for the observed effects requires more detailed studies.

In addition to uncovering functions of distinct classes and subtypes of cellular glycans, we also shed light on the requirement of viral glycans for proper function. HSV-1 particles are shown to contain both N-linked and O-linked glycans, and it is expected that GSL are part of the cell membrane-derived viral envelope[15–18]. It is not known whether proteoglycans decorated with GAGs can be incorporated into HSV-1 particles, but they were not detected in a proteomic study looking at cellular proteins incorporated into HSV-1 membranes[71]. While we identified ample functions of the host cellular glycans for HSV-1 biology, we found limited effects of glycome perturbations on properties of progeny virions able to establish infection.

It is important to consider the broader effects of glycogene knock out on cellular pathways and interaction networks. We have previously assessed transcriptomic profiles of *MGAT1* KO, *C1GALT1* KO, and *B4GALT5* KO N/TERT keratinocytes[6], where diverse transcriptomic changes affecting cell signaling, adhesion, and differentiation were identified, some of which may have bearings for the viral infection. Moving forward, correlating such data will be an important resource in dissecting the mechanisms affecting viral infections in more detail.

By using a glycoengineered keratinocyte library, we show that different classes and subtypes of glycans play very selective roles in specific stages in HSV-1 life cycle. We conclude that through evolution HSV-1 developed an elaborate network of interactions with different cellular glycans to allow for efficient entry and propagation, which could be explored for glycan-targeted intervention strategies. Individual findings of this exploratory study shall be the scope of future investigations. We furthermore highlight a glycoengineered human cell platform as a versatile tool for illuminating the glycobiology of other viral, bacterial, and fungal microorganisms.

## Methods

### Cells and viruses

The HaCaT human keratinocyte WT and genetically engineered cell lines were grown in DMEM supplemented with 10% FBS (HyClone) and 4 mM L-glutamine. CHOZN GS -/- (glutamine synthetase KO; WT CHO (Sigma)) and genetically engineered cell lines[33] were grown in EX-CELL® CD CHO Fusion serum free media mixed 1:1 with BalanCD CHO growth media and supplemented with 2% L-glutamine in 50 mL TPP TubeSpin® Bioreactors. DMEM mixed 1:1 with Ham's F12 nutrient mix and supplemented with 10% FBS (Gibco) and 4 mM L-glutamine was used for adherent CHO cell culture. CHO-Nectin 1 (HveC), CHO-HVEM, and parental CHO K1 cell lines were kindly provided by Richard Longnecker and cultured as stated above. HEK293 Ac2 cells were grown in DMEM supplemented with 10% FBS (Gibco) and 4 mM L-glutamine. The wild-type HSV-1 virus Syn17 + [72] was used throughout the study, and the virus titers were determined by plaque titration on Green monkey kidney (GMK, obtained from the Swedish Institute for Infectious Disease Control, Stockholm) cells as previously described[73]. 10-fold serial dilutions of viral stocks diluted in DMEM supplemented with 2.5% FCS and 100 unit/mL penicillin and 100 μg/mL streptomycin were allowed to adsorb to cell monolayers for 1 h at 37 °C with occasional agitation. The inoculum was then removed, and cells overlain with 2% methylcellulose in HBSS, mixed 1:1 (v/v) with DMEM supplemented with 5% FCS, 200 unit/mL penicillin and 200 μg/mL streptomycin. Overlain monolayers were incubated at 37 °C and 5% $CO_2$ for 48 h, overlay media removed, cells fixed and stained with 1% crystal violet in 70% EtOH/37% formaldehyde/acetic acid (20:2:1 v/v/v), washed 4 times with MQ $H_2O$, dried, and plaques counted using a light microscope (Olympus IMT-2). HSV-1 Syn17+ virus was cultivated in HaCaT wild type keratinocytes or GMK cells depending on downstream application and the titers were determined as mentioned above. K26GFP HSV-1[74] was a kind gift from Prashant Desai (The Johns Hopkins University), and cultivated as mentioned above.

### ZFN and CRISPR/Cas9 gene targeting

HaCaT KO library was generated using ZFN nuclease and CRISPR/Cas9 technology by targeting particular glycogene exons by validated gRNAs[75] or gRNAs predicted by GPP[76] (Supplementary Table 1, Supplementary Table 5). gRNAs were cloned[77] using oligos (TAGC, Denmark) into lentiCRISPR-v2-Puro plasmid backbone (Addgene #52961) or lentiCRISPR-v2-Blast with a blasticidin resistance gene replacing the puromycin resistance gene (Brakebusch laboratory, BRIC, UCPH, DK). Directional cloning and insertion of the gRNA duplex using BsmBI and T4 ligase into the LentiCRISPR-v2 plasmid backbone was done as described earlier[78]. All plasmids were propagated in One Shot™ Stbl3™ Chemically Competent *E. coli* cells (Thermo Fisher). Endonuclease free

plasmid preparations were made using Midi Prep kit (Thermo Fisher). For lentivirus production HEK293T cells were seeded at $1 \times 10^5$ cells/well density in a 6-well plate and grown for 72 h until 80–90% confluence. For transfection, 200 μL OPTI-MEM (Gibco), 8 μL of 1 mg/mL PEI (Sigma), 0.8 μg LentiCRISPR-V2-gRNA plasmid, 0.6 μg pCMV-VSV-G plasmid (Addgene #8454), and 0.6 μg psPAX2 plasmid (Addgene #12260) were mixed and incubated for 10 min at RT before added to the adherent HEK293T cells. After 24 h the transfection media was replaced with HaCaT media for harvesting virus for transduction. Medium containing viral particles was collected 48/72 h post-transfection, i.e., when virus had accumulated for 24 h after medium change, filtered (0.45 μm pore size). Filtered virus-containing medium was mixed 1:1 with fresh HaCaT media and 1:1000 polybrene (Sigma), and used to transduce HaCaT cells overnight[79]. Selection of KO cell lines started 48–96 h after transduction, with either 5 μg/ml blasticidin S (Gibco) or 1 μg/ml puromycin (Gibco) including biweekly cell passaging. Single clones were obtained by serial dilution in 96 well plates and KO clones were identified by IDAA using ABI PRISM™ 3010 Genetic Analyzer (Thermo Fisher) and Sanger sequencing (GATC, Germany). Two to five clones were selected for each gene with out of frame indel formation. IDAA results were analyzed using Peak Scanner Software V1.0 (Thermo Fisher)[80].

### MS-based glycan profiles

For O-GalNAc profiles, the Cellular O-Glycome Reporter/Amplification (CORA) method was used as described in[81]. In brief, HaCaT cells were seeded in 6-well plates, and when the cells reached 70% confluence, the medium was supplemented with 100 μM Bn-α-GalNAc or DMSO for untreated controls. Media was harvested after 48 h and passed through a 10-kDa Omega membrane (Pall Corporation) followed by purification with a Sep-Pak 3-cc C18 cartridge (Waters). Samples were then concentrated by SpeedVac to remove organic solvent and lyophilized. For profiling of N-linked glycans, HaCaT cell pellets were dissolved in 0.1% Rapigest (Waters) and incubated on ice and vortexed every 5 min. Samples were then centrifuged at 10,000 g for 10 min and the supernatant was collected and reduced with 5 mM DTT (Sigma) for 1 h at 60 °C followed by alkylation with iodoacetamide (Sigma) for 30 min at RT while kept dark. Samples were digested with 5 μg trypsin (Roche Diagnostics) overnight at 37 °C and peptides were purified with a Sep-Pak 1-cc C18 cartridge (Waters). Organic solvent was removed by SpeedVac and pH was adjusted to 7–8 with 50 mM ammonium bicarbonate, followed by overnight digestion with 3 U PNGase F (Roche diagnostics) at 37 °C. Released N-glycans were cleaned up on a Sep-Pak 1-cc C18 cartridge and lyophilized. For N-glycan profiles, dried samples were permethylated as described previously[82] and analysed by positive reflector mode MALDI-TOF (AutoFlex Speed, Bruker Daltonics) with data acquisition in the 1000–5000 m/z range. For O-glycans the data acquisition was in the 500–3500 m/z range.

### Disaccharide analysis

Disaccharide analysis was performed as previously described[33,36]. Approximately 20 million WT and *B4GALT7* KO HaCaT cells grown to confluency were harvested, washed in PBS, and resuspended in 50 mM Tris-HCl pH 7.4, 10 mM $CaCl_2$, 0.1% Triton X-100. 1 mg/mL pronase protease (Roche) was added and samples were incubated overnight at 37 °C. The pronase was heat inactivated at 98 °C for 10 min, after which Benzonase nuclease (Sigma-Aldrich) (250 U) and $MgCl_2$ (2 mM) was added for digestion of DNA and RNA at 37 °C for 2 h. HS and CS GAGs were then extracted from the samples through anion exchange, where samples were acidified to pH 4.5 by addition of acetic acid before being loaded onto freshly packed DEAE columns (0.5 ml beads, Sigma-Aldrich) equilibrated with 20 mM NaOAc, 100 mM NaCl, 0.1% Triton X-100, pH 5. Bound GAGs were eluted with 20 mM NaOAc, 1.25 M NaCl, pH 5, and subsequently precipitated by mixing elutes with NaOAc-saturated 100% ethanol (1:3 vol/vol). After centrifugation of samples at 20,000 × g for

20 min, the supernatant was discarded and the GAG containing pellets were dried by SpeedVac. GAGs were then dissolved in deionized water, and 1/2 of the total sample was used for digestion by chondroitinase ABC (Sigma-Aldrich) at 37 °C overnight, and 1/2 for digestion with heparinases I, II, and III (IBEX Pharmaceuticals) at 37 °C for 8 h adding first heparinase I, then heparinase III, and lastly heparinase II with 2 h intervals. CS GAGs were digested in a mix of 40 mM NaOAc pH 7, 1 μM CaCl$_2$, and 10 mU chondroitinase ABC, and HS was digested using 50 mM NaOAc pH 6.5, 5 mM CaOAc and 15 mU of each heparinase. After heat inactivation for 10 min at 98 °C, reactions were lyophilized and subsequently labeled with 2-aminoacridone (AMAC) by resuspension of pellets in 10 μL 0.1 M AMAC in 3:17 (vol/vol) acetic acid/DMSO. After 15 min incubation at RT, 10 μL of 1 M NaCNBH$_3$ was added followed by incubation at 45 °C for 3 h. Labeled disaccharides were then lyophilized and purified from excess AMAC twice by the addition of 500 μL acetone and centrifugation at 20,000 × g for 20 min, discarding the supernatant. Disaccharides were then dissolved in 2% acetonitrile before analysis on a Waters Acquity UPLC system with a BEH C18 column (2.1 × 150 mm, 1.7 μm, Waters), detecting fluorescence at 525 nm. CS disaccharides were separated by using 80 mM ammonium acetate pH 5.5 as mobile phase A and acetonitrile as mobile phase B, with mobile phase B increasing from 3 to 13% at a flow rate of 0.2 ml/min over 30 min. HS disaccharides were analyzed using 150 mM ammonium acetate, 100 mM dibutylamine, pH 5.6, as mobile phase A and 120 mM ammonium acetate, 80 mM dibutylamine and 20% acetonitrile, pH 5.6, as mobile phase B, with mobile phase B increasing from 18.5 to 100% at a flow rate of 0.2 mL/min over 36 min. CS and HS corresponding to 7.5 million cells were injected in each run. Disaccharide standards were purchased from Iduron, except 3-O-sulfated HS standards, which were purified from genetically engineered CHO cells. 20 picomoles of each standard was injected immediately prior to samples for identification and quantification of disaccharides.

### Organotypic culture

Organotypic cultures were prepared as described by Dabelsteen et al. [39,83]. Briefly, human fibroblasts were suspended in acid-extracted type I collagen (4 mg/mL) and allowed to gel over a 1-mL layer of acellular collagen in six-well culture inserts with 3-μm-pore polycarbonate filters (BD Biosciences NJ, USA). Gels were allowed to contract for 4–5 days before seeding with 3 × 10$^5$ HaCaT keratinocytes in serum and mitogen-supplemented DMEM/F12 raft medium. Inserts were raised to the air-liquid interface 4 days after cell seeding, and the media changed every second day for an additional 10 days. Infection of organotypic cultures was performed as previously described with minor modifications [84]. Briefly, media was removed on day 9 after insert raising, and 3000 PFU of HSV-1 Syn 17+ applied in 500 μL of RAFT media to the to the top of the cultures, let drip through in 15 min, and reapplied 3 more times during 1 h. RAFT media was then added to cover the bottom of the organotypic cultures and incubated for 36 more hours until harvesting.

### Propagation assay

HaCaT cell monolayers in 6 wells in triplicates were washed with PBS and infected with MOI 10 of HSV-1 Syn 17+. The virions in 1 mL of DMEM P/S were adsorbed at 37 °C with periodical rocking of the plates for 1 h, inoculum removed and cells washed with PBS, followed by addition of 2 mL/well of DMEM supplemented with 10% FCS and P/S. Virus was harvested at 17 h post infection. Media was spun at 3000 rpm for 10 min at 4 °C, supernatant aliquoted and stored at −80 °C. Cell monolayers were washed with PBS, dissociated with TrypLE reagent, spun at 100 × g for 5 min, washed with PBS, aliquoted and spun at 400 × g for 5 min, supernatant removed and cells stored at −80 °C.

### Plaque assay

Titers of virus produced in HaCaT keratinocytes were determined on GMK cells. Cell monolayers were infected with serial dilutions of virus and allowed to attach. After 1 h the inoculum was removed and the cells overlaid with medium containing 1% methylcellulose (Sigma-Aldrich), 2.5% FCS, 100 unit/mL penicillin and 100 μg/mL streptomycin (in HBSS (Sigma-Aldrich) + DMEM (Gibco, Life Technologies) at a ratio of 1:1). After 48 h incubation, the overlay medium was removed, the cells fixed with 1% crystal violet (in 70% EtOH/37% formaldehyde/acetic acid 20:2:1 v/v/v), washed 4x times with water and allowed to dry. The resulting plaques were inspected and counted using a light microscope (Olympus IMT-2). Plaque size was measured using AxioVision software.

### DNA extraction and qPCR

DNA was extracted using the NucleoSpin® Tissue Kit (Macherey-Nagel) according to manufacturer's instructions. Media was diluted 200x before extraction and 100 μL used for extraction. DNA was eluted in 100 μL volume. For assessing the DNA copy number of HSV-1, a 118-nucleotide segment of the gB-1 region was amplified with primers described in [85]. The qPCR reaction volume was set to 20 μL and contained 4 μL HOT FIREPol®Probe qPCR Mix Plus (ROX) (Solis Biodyne), primers and probe (forward primer at 0.5 μM, reverse primer at 0.5 μM and probe at 0.3 μM final concentrations), and 1 μl of DNA. Amplification of the target sequence was performed using the StepOne Plus system (Thermo Scientific). The reaction conditions were set to 10 min at 95 °C followed by 45 PCR cycles of two-step amplification (15 s at 95 °C and 60 s at 60 °C). HSV-1 Forward 5′-GCAGTTTACGTACAACCACATACAGC-3′; HSV-1 Reverse 5′-AGCTTGCGGGCCTCGTT-3′; HSV-1 Probe FAM-5′-CGGCCCAACATATCGTTGACATGGC-3′-MGBNFQ (Thermo Fisher Scientific). The efficiency of each round of PCR was determined using 10-fold dilutions of Topo TA plasmids (Invitrogen AB, Stockholm, Sweden) with insert of respective amplicon created according to the manufacturer's instructions.

### Binding and entry assays

Binding and entry assays were performed as previously described [86]. HaCaT cell monolayers were prechilled on ice for 30 min. For binding assay, 200 PFU (MOI < 0.0005) of HSV-1 Syn 17 + WT or glycoengineered HSV-1 in 1 mL DMEM supplemented with 2.5% FCS, 100 unit/mL penicillin and 100 μg/mL streptomycin (P/S) was added on cell monolayers on ice in triplicates, and incubated for 20 or 120 min. After the indicated time points, cells were washed twice with ice-cold PBS and overlain with 2% methylcellulose in HBSS, mixed 1:1 (v/v) with DMEM supplemented with 5% FCS, 200 unit/mL penicillin and 200 μg/mL streptomycin. Overlain monolayers were incubated at 37 °C and 5% CO$_2$ for 48 h, overlay media removed, cells fixed and stained with 1% crystal violet in 70% EtOH/37 % formaldehyde/acetic acid (20:2:1 v/v/v), washed 4 times with MQ H$_2$O, dried, and plaques counted using a light microscope (Olympus IMT-2). For entry assay, after 120 min on ice, the inoculum was removed, washed with room temperature DMEM, P/S, and 1 mL of room temperature DMEM supplemented with 2.5% FCS and P/S was added, followed by a 5 min incubation at 37 °C. The plates were then returned on ice, media removed and non-entered particles inactivated with ice-cold citrate buffer (40 mM citric acid, 10 mM KCl, 135 mM NaCl pH 3) for 2 min, then washed twice with ice-cold PBS and overlain with media as described above. A 0 min time point without shift to 37 °C was used as control.

### Antibodies and probes

Mouse anti-gB (HSV-1B11D8; 1:20 used for CELISA, 1:50 for imaging), anti-gC (B1C1B4; 5 μg/mL used for imaging, 2 μg/mL for flow cytometry), anti-gD (C4D5G2; 1:100 used for CELISA and for imaging) and anti-gE (B1E6A5; 1:100 for imaging) mAbs, as well as recombinant gC (2 μg/mL used for imaging, 0.5 μg/mL for flow cytometry) probe were

produced at the University of Gothenburg[87]. Shiga toxin 1, B subunit (StxB) was purchased from Sigma (Cat. Nr. SML0562) and labeled using NHS-Fluorescein (5/6-carboxyfluorescein succinimidyl ester, Thermo Scientific Cat. Nr. 46410) followed by dye removal using ZEBA Spin 7 MWCO desalting column (Thermo Scientific); used 1:4000 for imaging. Rabbit anti-E-cadherin mAb (24E10; 1:200 for imaging) was purchased from Cell Signaling Technology and goat anti-HSV-1-FITC pAb (GTX40437; 1:50 for imaging) from Genetex. Mouse anti-Nectin 1 mAb (clone CK8, Cat. Nr. 37-5900; 1:50 for flow cytometry) and rabbit anti-HVEM pAb (Cat. Nr. PA5-29780; 1:50 for flow cytometry) were purchased from Thermo Scientific. Mouse anti-ceramide mAb (Cat. Nr. MAB_0010; 1:1000 for imaging) and rabbit anti-glucosylceramide antiserum (Cat. Nr. RAS_0010; 1:400 for imaging) were from Glycobiotech, and mouse anti-GM3 (clone CGYJ074; 1:100 for imaging) mAb was from Creative Biolabs. Goat anti-mouse IgG AF647 F(ab)$_2$ (1:500 for imaging), goat anti-mouse IgG AF594 (H + L) (1:500 for imaging), goat anti-mouse IgG AF488 (H + L) (1:500 for imaging, 1:1000 for flow cytometry), goat anti-mouse IgM AF546 (H chain) (1:500 for imaging), goat anti-rabbit IgG AF488 (H + L) (1:500 for imaging, 1:1000 for flow cytometry), and donkey anti-rabbit IgG AF546 (H + L) (1:500 for imaging) were all from Thermo Fisher Scientific. Rabbit anti-mouse Igs HRP was from DAKO (1:500 for CELISA).

## Immunofluorescence

HaCaT cells grown on glass cover slips in 24-wells were infected with MOI 10 of HSV-1 K26-GFP. 14 h post-infection the cells were washed 2x with Hanks' Balanced Salt solution (HBSS, Sigma-Aldrich Cat. Nr. H8264) and fixed with 4% PFA in PBS (Ampliqon Cat. Nr. AMPQ44154.1000) for 15 min at RT followed by 2x more washes. The cells were permeabilized with 0.3% Triton X-100 in HBSS for 3 min followed by 2x washes with HBSS and blocked with 2.5% BSA in HBSS, 0.03% sodium azide at RT for 45 min. Cover slips were incubated with primary antibodies diluted in blocking buffer at 4 °C over night followed by 3x washes with HBSS and incubation with secondary antibodies at RT for 45 min. The cover slips were rinsed with HBSS and incubated with 0.1 μg/mL DAPI solution in HBSS for 5 min followed by 3x washes and mounting with Prolong Gold antifade-reagent (Thermo Fisher Scientific Cat. Nr. P36930). For stainings of non-infected HaCaT cells with gC probe, fixed and permeabilized cells on cover slips were treated with either chondroitinase ABC (0.25 mU), heparinases 1, 2 and 3 (0.25 mU of each), or left untreated for 1 h at 37 °C followed by incubation with 2 μg/mL gC probe diluted in 2.5% BSA in PBS, 0.03% sodium azide for 1 h at RT, 3x washes with PBS and staining with mouse anti-gC mAb as described above, though using PBS-based blocking buffer and washes. For plaque stainings HaCaT cells grown on large cover slips in 6 wells were infected with 200 PFU (MOI < 0.0005) of HSV-1 K26-GFP and overlaid with DMEM, 2.5% FCS, 1% methylcellulose, 100 unit/mL penicillin, 100 μg/mL streptomycin for 48 h. Therafter the overlay media was removed, cover slips washed 3x with PBS and fixed in 4% PFA in PBS for 25 min followed by 3x washes and blocking in 2.5% BSA in PBS, 0.03% sodium azide at 4 °C over night and staining procedure as described above, though with primary Ab incubation for 2 h at RT and dilutions in 2.5% BSA in PBS, 0.03% sodium azide. Images were documented using Zeiss LSM710 confocal microscope using Zen Black 2012 SP5 Software. FFPE organotypic tissue sections were deparaffinized and microwave treated for antigen retrieval (Tris-HCl buffer pH 9). The sections were then permeabilized with 0.3% Triton X-100 in PBS, washed 3x and incubated with goat anti-HSV-1-FITC pAb (GeneTex, 1:50 in 2.5% BSA in PBS, 0.03% NaN$_3$) for 2 h at room temperature, followed by 3 washes with PBS and mounting with ProLong™ Gold antifade reagent with DAPI (Thermo Fisher Scientific). Entire sections were imaged using Zeiss AxioScan microscope. Images were assembled using Adobe Photoshop 2021 (version 22.3.0), Adobe Illustrator 2021 (version 25.2.1), or Zeiss ZEN Lite (version 3.4.91.00000) software.

## Flow cytometry

Cell surface binding studies were carried out by flow cytometry analysis. For suspension CHO cell analysis, cells were counted and $1 \times 10^5$ cells/well were distributed into U-bottom 96-well plate (TermoFisher Scientific™ Nunclon™ Delta surface). Cell viability was evaluated by trypan blue stain, applying a viability requirement of >95% for cells to be included for analysis. Next, cells were washed once in ice-cold PBS, centrifuged at 500 × g for 3 min at 4 °C. Cells were then incubated with recombinant gC protein (0.5 μg/mL) diluted in ice-cold PBS with 2% FBS, for 45 min at 4 °C. Following, cells were washed twice in ice-cold PBS with 2% FBS before incubation with mouse anti-gC mAb B1C1B4 (2 μg/mL) diluted in PBS with 2% FBS for 45 min at 4 °C. After incubation, cells were washed twice in ice-cold PBS with 2% FBS and incubated with goat anti-mouse IgG (H + L) AF488 (1:1000) diluted in PBS with 2% FBS for 45 min at 4 °C in the dark. Lastly, cells were washed twice with PBS with 2% FBS, and suspended in 100 μL PBS with 2% FBS for analysis. Fluorescence intensity was measured on the SONY Spectral Cell Analyzer (SONY SA3800). A total minimum of 10,000 events were recorded for analysis. Cell debris and doublets were excluded based on SSC-A/SSC-H plot to ensure analysis based on singlets (Supplementary Fig. 3). Data is presented as the geometric mean of samples incubated with gC probe, after subtraction of the geometric mean of background fluorescence, samples incubated with primary and secondary antibody only, and normalized to WT. For the analysis of HaCaT, CHO K1, CHO-Nectin 1 (HveC), CHO-HVEM, and HEK293 cells, $4 \times 10^5$ cells/well were used. In addition, a cell viability stain (eBioscience™ Fixable Viability Dye eFluor™ 450, 1:1000 in PBS) for 30 min at 4 °C prior to antibody labeling was included. The cells were then incubated with relevant primary (mouse anti-Nectin 1 1:50 or rabbit anti-HVEM 1:50 in 1% BSA in PBS) and subsequent secondary (goat anti-mouse IgG AF488 (H + L) or goat anti-rabbit IgG AF488 (H + L) 1:1000 in 1% BSA in PBS) Abs for 30 min at 4 °C each. Cells were washed with 1% BSA in PBS thrice between all the incubation steps. Incubation with secondary antibodies only were used for background subtraction. Cell debris and doublets were excluded based on FSC-A/FSC-H to ensure analysis based on singlets, and dead cells were excluded based on the viability dye stain (Supplementary Fig. 3). All data analysis was done using FlowJo V. 10.0.7.

## Plasmids and cloning

Plasmids encoding full length HSV-1 gB (pMTS-1-gB), gD (pcDNA3.1(-)-gD), gH (pMTS-1-gH), and gL (pMTS-1-gL) was a kind gift from Gabriella Campadelli-Fiume (University of Bologna). Plasmids encoding split luciferase (DSP1-7, DSP8-11) were kindly provided by Zene Matsuda (University of Tokyo). Single and multiple amino acid mutations in gB and gD were introduced using custom primers (Supplementary Table 4) and the Quik-Change Lightning Multi Site-Directed Mutagenesis Kit according to manufacturer's instructions. Briefly, mutagenic primers were annealed and extended during 30 thermal cycles (2 min 95 °C; 30 x (20 s 95 °C, 30 s 55 °C, 30 s/kb 65 °C); 10 min 65 °C), followed by template digestion with DpnI, and transformation into XL10-Gold Ultracompetent Cells. The transformation reactions were plated on LB-carbenicillin agar plates and colonies screened by PCR using Herculase II Fusion Pfu DNA Polymerase (Agilent) followed by Sanger sequencing. Colonies of interest were re-streaked, grown up in LB-carbenicillin media and Mini Preps prepared using Zyppy Plasmid Miniprep Kit (Zymo Research) followed by Sanger sequencing. Subsequently, plasmid Midi Preps were prepared using NucleoBond Xtra Midi Endotoxin-free Kit (Macherey-Nagel).

## Split-luciferase assay

The assay was performed as previously described[40] with several modifications. CHO K1 cells (effector) grown in Greiner CELLSTAR® white polystyrene 96 well plates were transfected with 375 ng/well of

gB, as well as 125 ng of each gD, gH, gL, and DSP1-7 plasmids using Lipofectamine 3000 (Thermo Scientific) according to manufacturer's instructions in triplicates. Plasmids encoding wild type or mutated gB or gD were used. An additional clear polystyrene plate with transfected cells was prepared for CELISA measurements. HEK293, CHO-Nectin 1 (HveC), or CHO-HVEM cells (target) grown in 6 well plates were transfected with 1 µg/well of DSP8-11 plasmid. For split-luciferase assay, 23 h post-transfection, the media of CHO K1 cells was replaced with fusion media (DMEM (no phenol red), 5% FCS, 50 mM HEPES), supplemented with EnduRen substrate (Promega, 1:1000) and incubated at 37 °C for 1 h. Target cells were detached using 0.02% EDTA in PBS, resuspended in fusion media and added to effector cells. Luminescence was measured every 2 min for up to 240 min at 37 °C using EnSpire plate reader (Perkin-Elmer).

## CELISA
Cell surface ELISA was performed as previously described[40] with several modifications. 96w plates with transfected cells were washed 3x with PBS supplemented with divalent cations (0.9 mM $CaCl_2$, 0.5 mM $MgCl_2$ ($PBS^{2+}$)) 20 h post-transfection and fixed with 4% PFA in PBS for 10 min followed by 3x washes and blocking in 3% BSA in $PBS^{2+}$ for 30 min. The cells were then incubated with mouse anti-gB (1:20) or anti-gD (1:100) mAbs in blocking buffer for 1 h at room temperature, followed by 5x washes and incubation with rabbit anti-mouse Igs HRP (1:500) for 45 min and 6x washes. Plates were developed using TMB + substrate chromogen (Thermo Scientific) and stopped with 0.5 M $H_2SO_4$. The absorbance at 450 nm was quantified using BioTek Synergy LX plate reader.

## Statistical analysis and data visualization
Prism 9 software version 9.5.0 was used for generating graphs and for performing statistical analyses. PyMOL software version 2.1 was used for generating protein structure illustrations using publicly available protein structure coordinates.

## Reporting summary
Further information on research design is available in the Nature Portfolio Reporting Summary linked to this article.

## Data availability
All data generated or analysed during this study are included in this published article (and its supplementary information files). Source data are provided with this paper. Protein structure coordinate files (PDB: 2GUM and PDB: 2C36) were downloaded from the Protein Data Bank for generating protein structure illustrations in PyMOL. Source data are provided with this paper.

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

## Acknowledgements

We thank Karin Uch Hansen for technical assistance. We acknowledge the Core Facility for Integrated Microscopy, Faculty of Health and Medical Sciences, University of Copenhagen. We kindly thank Prashant Desai, Zene Matsuda, Gabriella Campadelli-Fiume, and Richard Longnecker for sharing virus strains, plasmids, and cell lines. We also thank Gary Cohen and Doina Atanasiu for their valuable guidance. This work was supported by Lundbeck Foundation (R219-2016-545, IB), Danish National Research Foundation (DNRF107) and European Comission (GlycoSkin H2020-ERC, HHW).

## Author contributions

Conceptualization, IB and HHW; Methodology, IB, SO, RN, TB Validation, IB; Formal Analysis, IB, INM, AMRL, RK, and RLM; Investigation, IB, INM, AMRL, EMHP, SLKS, RK, TBR, SD and HHW; Resources, YHC, SO, RN, TB and HHW; Data Curation, IB; Writing—Original Draft Preparation, IB and HHW; Writing—Review & Editing IB, AMRL, TBR, YHC, RLM, SO, RN, TB, and HHW; Visualization, IB, INM, and HHW; Supervision, IB and HHW; Project Administration, IB and HHW; Funding Acquisition, IB and HHW. All authors have read and agreed to the published version of the manuscript.

## Competing interests

Unrelated to the presented work, HHW owns stocks and is a consultant for and co-founder of EbuMab, ApS, Hemab, ApS, and GO-Therapeutics, Inc. All other authors declare no competing interests.
