## [Peer Review File · Nature Communications]

Glycoengineered keratinocyte library reveals essential functions of specific glycans for all stages of HSV-1 infectionREVIEWER COMMENTS

Reviewer #1 (Remarks to the Author):

The role of glycans in the different steps of viral lifecycles is a very important topic which has been understudied over the years. One of the reasons for the lack of information about that is the difficulty to study a system which is non-template driven. The recent advances of mass spectrometry are progressively filling this gap but there is still a lot to do to develop accessible systems to address those questions in a meaningful way.

In the present study, Bagdonaite et al. developed an elegant system to address the importance of different glycans in the biology of human Herpes Simplex 1 virus (HSV-1). The authors generated glycoengineered HaCaT keratinocytes by introducing targeted deletions in glycosyltransferase genes resulting in glycan changes at initiation, branching and capping events, and encompassing all the major classes of glycoconjugates.

The approach is innovative and very interesting. Moreover, some parts of the work, such as the part on GAG binding (figure 2 and especially panels F-H), are very convincing. However, at many places, the way to present the results (ratio, ratio of ratio, ...) and the absence of statistically significant differences in most of the comparisons make it very difficult to analyse the results. It therefore seems that it requires deep re-analysis and/or additional repetitions in order to fully convince readers about the differences that are shown.

Major comments

- In general, the statistics applied to the panels B to D of figure 1 are not clear. Indeed, from lines 138 to 162, the authors make hypotheses about results that are sometimes not statistically significant (based on the arrow on the figures). For some (ie Core 1 O glycans) this looks clear enough (even if surprisingly not significant) but for others, this is not clear. The presentation of results as ratio of absolute values that are probably in log scales make it very difficult to interpret.
- The same comments apply to figure 2. How to be sure that there is a difference when this is not statistically significant? For example, with the ST3GAL1 KO, there is a tendency but this is not significant based on the statistics shown. Therefore, even if the approach is very innovative and nice, how to conclude about something?
- In Fig. 2D, is the Y axis legend ok? Is it WT/KO or KO/WT?
- The staining presented in Figure 3 are nice. However, how to reconcile those results with the ones obtained in figure 1? Again, what is significant and what is not?
- In Fig. 3C, how to explain the weaker signal at 20h pi than at 14h pi?
- In figure 4, the normalization relative to protein surface expression does not appear so robust. Indeed, surface expression signal depends on a large amount of proteins whereas, the success of fusion is revealed similarly when one or thousands of fusion events occurred per cell. This should be clarified. One solution would be to make the assay with dilutions of the same plasmid and then see if reduced cell surface expression translates in differences in fusion readout.
- What is the rationale to do it with gD in figure 4? As mentioned by the authors gD does not mediate fusion. However, the readout is fusion.
- In Figure 5, when the authors suspect an effect of the mutations on the glycoproteins glycosylation (for example with gE N glycans), they could show it by western blot in order to support their hypotheses. This would be nice to see the molecular weight of gE in the different KO cells treated or not with specific enzymes removing glycans.
- The comparison of 2D with 3D cultures is very interesting, however, the results do not appear coherent. The technique applied to figure 5D-G is very nice, however the interpretation is difficult as, here too, most of the differences observed are not significant. This should be investigated further.

Minor comments

- The legend of Figure 1 is not correct. It seems that there is a mix with Figures S1 and S2.
- Before comparing viral growth on the KO cell lines, it could have been interesting to show that mutations do not affect cell survival or proliferation as it could have direct consequences on virus

growth.

- Panels B-D of Figure 1 appear very interesting but they are always presented as ratio or ratio of ratio (panel D). It would have been interesting at some point to have an idea about the absolute number of PFU or viral genome copies.
- The decreased fitness suggested for virions produced on MGAT1 KO cell seems to be non significant. The text should therefore be changed.
- The sentence in lines 165-166 is not clear. Indeed, how can it be said that binding is complete or not?
- Lines 389-390, could the author clarify the sentence: "where dissociation of progeny virions is restricted"

Reviewer #2 (Remarks to the Author):

Authors are correct in stating that there are limited accessible research tools and model system to cellular and viral glycan biology. Within such context, glyco-engineered cells are welcome. However, with multiple different cell types in the human body, the 'right choice' of cells to have the glyco-engineering would attract the most attention from the community (in my opinion).

Authors are also correct stating that glycans 'being the most diverse post-translational modification', glycan dynamically shaping the glycans on proteins and lipids through the secretory pathway. It is with these recognised challenges; authors perhaps need to control with their experiments much more tightly and / or tone down their claims. More specifically, the domino effect on any glycosylation related enzyme manipulation will impact on the target cells in terms of natural biology.

Specifically, any of these glycan modification in N-glycan or O-glycan will give rise to a series of domino effects that are not just related to viral proteins, but ALL cellular proteins rely on these specific glycosylation events. As authors pointed out, these wide spectra of glycosylated proteins will contribute to multiple series and layers of ripple effects on these viral and cellular glycol-conjugates to influence their biological processes. Consequently, virus titter and viral genome DNA copies numbers are blunt instruments to reveal specific biological impacts. It would have been much nicer if the authors can have examined a specific biological process relates to HSV-1, go deeper into specific virological events.

The data outcome (claims) within the section in 'Binding and entry HSV-1 to mutant cell lines' are expected. In my opinion, one might have missed the opportunity to use these specific glyco-engineering to dissect the mechanisms better. In my opinion, current presentation might not be the best applications / usages of these really cool glycoengineered cells. With ALL glycan molecules on targeted cells are affected by a single glyco-related enzyme, it is extremely difficult to understand the precise biological impacts.

It seems to me that the authors are much stronger glyco-biologists than virologists, and perhaps the pitch and the focus of the story should be presented as such.

In its current form, the manuscript appears more like to potpourri of some nice glycobiology observations with selected glycoengineered cells without a highly coordinated story for a high-impact journal of this calibrate. For example, using each one of these glyco-engineering cell to go deep and dissect a specific aspect of the story would be really nice.

The science is great, but the data appear to be disjoint (to me) in terms of storytelling. In current form, it feels more like a series of 'capable experiments' by a strong research team.

For example, while the platform of constructing these glycoengineered cells are highly valuable, but if the host cells are only with keratinocytes, the application with virology will be limited. Similarly, if the idea is to have glycoengineered cells to study biological processes, this assessor

would like to see the platforms to be applied to other relevant primary cells that can accelerate biological and scientific discoveries in the context of relevant glyco-biological questions. Other mucosal cells or immune cells that rely on glycobiology to interact with pathogens would be excellent addition.

The domino effects of these glyco-engineering cells will always be a concern. Perhaps coupling with some mix-omics analyses on some of the relevant cell types would suit the purpose for platform for the field.

This assessor (personally) feel that there are many good parts in this manuscript that can be further developed to be high impacts paper on their own. In current form, it feels (to me) that the aspects being presented do not 'gel well' for this audience.

Reviewer #3 (Remarks to the Author):

Glycans are known to be exploited by multiple human pathogens. In this study authors reports the glycoengineering of HaCaT keratinocytes by introducing targeted deletions in the cellular glycome, including select initiation, branching and capping events, covering all the major classes of glycoconjugates to delineate human glycosylation pathways for their potential role for specific glycans in the infectious life cycle of herpes simplex virus type 1 (HSV-1). The study finds that GAGs and glycosphingolipids are important for HSV-1 attachment, while N-glycans play critical role for entry and spread, and O-glycans for needed for virus propagation. Further, it was found that the altered virion surface structures had minimal effects on the early interactions in the wild type cells, however the mutation of specific O-glycosylation sites affected glycoprotein surface expression and function. In conclusion, the study highlights the significance of specific glycans in a clinically relevant human model of HSV-1 infection and urges the further exploration of genetic engineering may further to elucidate novel antiviral interventions.

Certainly, the work is novel and has significance in the herpes entry field. The experiments and methodology along with the controls have been done carefully and manuscript should be considered for publication after clarification with some additional experiments for evidence to further strengthen the overall goals of this finding.

1. Since the glycans are known regulators as a house keeping gene, it is obvious to ask if the KO Keratinocytes library affected membrane topography with the GAG expression and potential cellular signaling ability. An assessment profiles for the above compared to control will be useful.
1. In fact, a control human herpesvirus (CMV or EBV) should have been used to reach to the conclusion (but it is not required).
2. In addition, multiple HSV-1 gD receptors are also tethered with the cell membrane, it is critical to evaluate if KO Keratinocytes (mutants) may have also affected their expression (nectin/HVEM) level. This can be shown either by cell-ELISA or flow cytometry.
3. The intriguing question remain unanswered that which gD receptors and internalization modes along with glycans mediate the HSV entry in vivo at the natural sites of virus infection.
4. It is unclear if the sulfated GAGs contribute only to viral attachment (as per the comments made in the manuscript), since 3-O sulfated-HS has been shown to generate as an independent gD receptor allowing virus-cell fusion.
5. 200 pfu needs to be expressed in terms of MOI, as some places the infections are represented as MOI.
6. The Fig. 1E, and 1F mentioned in the figure legends are missing.
7. Figure 4 B cell to cell fusion model is not well explained, why the presence of cell surface glycans in CHO and HEK293T cells have been ignored. What will be the type(s) of gD receptor in HEK293T cells?
8. Figure 4 D, E and H have two panels and therefore should be re-numbered.
9. Fig. 4 C why gB expression itself is affected? Please provide the stats.
10. Please fix the Fig. 5C, why upper (higher x) and lower panels (lower x) are not taken from the

same image.

11. Figure 5F, please mention green stands for HSVP26 GFP and blue stains for what?

12. Discussion section says – “We utilized a genetically engineered keratinocyte library to systematically investigate the contributions of different classes of glycan structures to multiple stages of HSV-1 life cycle in the natural context of infection”. Please be specific and clear that here you are talking about cellular glycans but not the viral glycans. Are viral and cellular glycans can have homotypic or heterotypic interactions as well.

13. Discussion section – “Since distinct glycan classes could independently finetune early virus-host interactions, we suggest that HSV-1 through its evolution developed elaborated mechanisms to exploit the host glycosylation machinery”.- what is basis of such statement, did you check the evolutionary herpes viruses that potentially affected aquatic species and had differences in terms of glycan selection?

14. In Discussion section – “We identified diverse effects on the infectious cycle attributable to select maturation or branching steps of protein-bound N- and O-linked glycans” – again be specific that here you are highlighting the significance of viral glycans.

15. Does all the clinical HSV strains presumably loaded with variable viral glycans could use similar cell glycan signature to enter the host cell, or the phenomenon could just be strain specific/cell specific is not clear. The clinical significance of glycoengineered viruses is not clear. It would have been great to use clinical isolates in this study.

16. Since the glycans are also involved both in the interaction of a prokaryotic bacterium or virus binding to a eukaryotic host cell, and the interactions between the cells of the immune system, nothing has been discussed in terms of nosocomial infection which could potentially utilize similar route. Therefore, the overall implications or significance of the current study may have propounding impact in the future.

17. Furthermore, Glycans have been shown to function as versatile molecular signals in cells and therefore it is not clear if blocking or targeting glycans would bring an unwanted effect on the host cell.

18. Lot of references are missing in the original statement throughout the manuscript.

Comments to reviewers

General comment

We thank the reviewers for their valuable feedback, which we carefully addressed by additional experimentation and data reanalysis, as outlined in the **point-by-point query-response list** below. On a more general note, the reviewers raised concerns about the conceptual advances provided by the study regarding HSV-1 biology and the unpredictable effects of glycoengineering on cell biology.

While various HSV-1 protein entry receptors have been identified in different cell types, functions of glycans, another type of highly abundant and diverse cell surface biomolecules have not been systematically studied in the cellular context due to difficulties in manipulating non-template driven synthesis of glycans, which is now achievable by state-of-the-art precise gene editing tools.

We would like to emphasize that we utilized a biologically relevant model to systematically investigate the roles of glycans in HSV-1 infection by generating a library of cells differing in their glycosylation capacity, and encompassing several biosynthetic steps of all major classes of human glycans. Compared to other cell types routinely used in viral research, such as monkey kidney cells (Vero) or baby hamster kidney cells (BHK), we used a human tissue-forming skin cell line representing the primary site of infection of HSV-1, where the virus undergoes its lytic cycle resulting in the common clinical manifestation of cold sores. Thus the findings of our study is highly relevant for understanding the contributions of glycans to HSV-1 pathology in humans.

Our additional work on characterizing the viral receptor and glycan landscape allowed us to identify Nectin 1 as the predominant entry mediator in HaCaT keratinocytes, with only minute amounts of HVEM or 3-O-sulphated HS. We further show that alterations in N-linked glycan maturation and branching, as well as O-linked glycan branching affected the surface availability of glycoprotein Nectin 1, which correlates with our observed effects on viral attachment efficiency. In contrast, changes in membrane bound glycolipids did not affect Nectin 1 surface expression, and yet markedly reduced viral entry. By further characterization of membrane glycolipid distribution, we show highly heterogeneous spatial distribution of GSLs, possibly contributing to receptor presentation.

Altogether, this suggests that entry into human keratinocytes is highly regulated and sensitive to moderate alterations in entry receptor availability and the context of the remaining glycocalyx.

We believe that with our platform we identify several key effects mediated by glycans and provide a first glance at viral tissue spread with and without specific glycans. We would like to once again stress that the findings of our genetic screen are selective to different stages of the viral infectious cycle and the types of glycoconjugates. The contribution of glycosphingolipids to HSV-1 entry is novel and opens a whole new line of research. Furthermore, our findings on specific GAG motifs and GSL structures holds promise for

development of tailored glycan-based inhibitors in the form of custom synthesized GAGs and glycolipid-enriched liposomes.

We acknowledge that altering total cellular glycomes can have unpredictable effects on cell signalling and other basic properties, and represents a limitation of the glycoengineering strategy, which needs to be addressed when investigating individual findings in future studies. However, we firmly believe that the benefits of the platform outweigh the drawbacks compared to out-of-context experimental systems.

We would like to state again that by applying a clinically relevant glycoengineered cell and tissue library with a live human pathogen we were able to identify several distinct and previously unknown contributions of glycans for a given virus, which has broad applicability to other human viruses and other types of microbes. We thus present a strategy allowing to identify critical steps within glycosylation pathways paving the way for future focused studies addressing the mechanisms involved. We agree that it would be relevant to also investigate other cell lines and clinical HSV-1 isolates, but we argue that our study will allow us and others to do so in a focused manner using different types of cell culture, e.g. of neural or immune origin.

Point-by-point query-response list

Reviewer #1 (Remarks to the Author):

The role of glycans in the different steps of viral lifecycles is a very important topic which has been understudied over the years. One of the reasons for the lack of information about that is the difficulty to study a system which is non-template driven. The recent advances of mass spectrometry are progressively filling this gap but there is still a lot to do to develop accessible systems to address those questions in a meaningful way.

In the present study, Bagdonaite et al. developed an elegant system to address the importance of different glycans in the biology of human Herpes Simplex 1 virus (HSV-1). The authors generated glycoengineered HaCaT keratinocytes by introducing targeted deletions in glycosyltransferase genes resulting in glycan changes at initiation, branching and capping events, and encompassing all the major classes of glycoconjugates.

The approach is innovative and very interesting. Moreover, some parts of the work, such as the part on GAG binding (figure 2 and especially panels F-H), are very convincing. However, at many places, the way to present the results (ratio, ratio of ratio, ...) and the absence of statistically significant differences in most of the comparisons make it very difficult to analyse the results. It therefore seems that it requires deep re-analysis and/or additional repetitions in order to fully convince readers about the differences that are shown.

Major comments

Remark 1. In general, the statistics applied to the panels B to D of figure 1 are not clear. Indeed, from lines 138 to 162, the authors make hypotheses about results that are sometimes not statistically significant (based on the arrow on the figures). For some (ie Core 1 O glycans) this looks clear enough (even if surprisingly not significant) but for others, this is

not clear. The presentation of results as ratio of absolute values that are probably in log scales make it very difficult to interpret.

Response 1. We considered different options how to present this data. To control for experiment-to-experiment variation in viral output, and because our experiments included up to 4 cell lines at a time, we found it useful to normalize to WT in each experiment, and also intuitive to view data in ratios. While useful for visualization, we realize it was not optimal to perform statistical analysis on normalized data, which limited the choice of tests and prevented us from confidently identifying meaningful differences. We also acknowledge that perhaps it would have been more informative to view actual titers/DNA copies.

We therefore took the advice to perform a deep reanalysis of our data. We have now restructured our non-normalized data in a “grouped” type analysis and then used a two-way ANOVA to analyze the differences, as this type of analysis takes into account the pairing of samples in different experiments. This more appropriate analysis provided us confidence in our conclusions. We now also show non-normalized data in Figure 1, and we edited the text throughout the manuscript to remove interpretations of non-significant data.

Action 1. We updated Fig. 1 and the text on page 7 (lines 143-156):

“b-d The array of cell lines was infected with MOI 10 of HSV-1 Syn17+, and viral titers **b** and viral DNA content **c** in the media were measured at 17 hpi. Particle to PFU ratio is also shown (**d**). Data points show average values \pm SEM from 3 independent experiments for each knock out (KO) cell line including paired WT data from a total of 14 experiments. Two-way ANOVA followed by Dunnett’s multiple comparison test was used to evaluate differences from WT (* $p < 0.05$, ** $p < 0.01$, *** $p < 0.001$, **** $p < 0.0001$). e-g Data from b-d is shown as WT-normalized mean \pm SEM of 3 independent experiments for each knock out (KO) cell line.”

Text edits:

“When infecting cells with truncated O-glycans (*C1GALT1* and *C1GALT1C1* KOs) a decrease in viral titers was detected (Fig. 1b, 1e). In contrast, the same cells generated close to normal

levels of viral DNA (Fig. 1c, 1f), suggesting decreased fitness of virions lacking elongated O-glycans (Fig. 1d, 1g). This feature was unique to complete truncation, and not seen when eliminating branching or sialylation of O-glycans. Looking into sialylated core 1 O-glycans, we found a trend towards accelerated production of viral particles in cells deficient in O-glycan sialylation (*ST3GAL1* KO). As the viral particles from these KO cells displayed normal fitness characteristics, the data indicate that each step of the O-glycan machinery has a distinct role in virus production and maturation. In cells lacking N-glycan maturation (*MGAT1* KO) we also found a lower number of infectious particles (Fig. 1b, 1e) with an apparent decreased fitness as indicated by an increase in the ratio of DNA/infectious particles (Fig. 1d, 1g). This apparent decrease in fitness was not detected in cells with loss of N-glycan branching, and in *MGAT4A* KO cells we even observed an overall increased viral output (Fig. 1b, 1c). Interestingly, this was different in cell lines lacking the β 6-antennae on N-glycans (*MGAT5*, *MGAT5+4B*) where we found that a higher proportion of progeny virions were infectious, suggesting increased fitness (Figure 1D). When analysing cells with glycosphingolipid synthesis defects, we found that lack of LacCer sialylation (*ST3GAL5* KO) accelerated virus production (Fig. 1b, 1c, 1e, 1f), but without any change in viral fitness (Fig. 1d, 1g). Finally, loss of cellular glycosaminoglycans (GAGs) increased the production of viral particles, though with a slight decrease in viral fitness (Fig. 1c). In conclusion, most of the tested glycosgene disruptions permitted HSV-1 replication, and only disruption of N- or O-glycan maturation impaired viral fitness.”

Remark 2. The same comments apply to figure 2. How to be sure that there is a difference when this is not statistically significant? For example, with the *ST3GAL1* KO, there is a tendency but this is not significant based on the statistics shown. Therefore, even if the approach is very innovative and nice, how to conclude about something?

Response 2. We again acknowledge that our choice of analysis was suboptimal. Similar to Figure 1, we now reorganized and reanalyzed all our raw non-normalized data using two-way ANOVA, allowing to reaffirm convincing differences with glycolipids, GAGs, and N-glycan branching. We still elected to present normalized data, but all raw data is included in a supplementary source data file. We also agree that we should be cautious about interpreting non-significant results. We therefore edited the text to remove such interpretations.

Action 2. We update Fig. 2a, 2b, and 2d based on 2-way ANOVA analysis of raw data, and edited the text on pages 7-8 (lines 163-164).

“Perturbations in each of the investigated glycosylation pathways modulated early virus-host interactions (Fig. 2a-e). **Cells lacking core 1 O-glycan sialylation (ST3GAL1 KO) showed reduced binding (Figure 2A, 2B), which was also reflected in the subsequent entry experiments (Figure 2D, 2E).”**

Remark 3. In Fig. 2D, is the Y axis legend ok? Is it WT/KO or KO/WT?

Action 3. Thank you for pointing out the error, which we now corrected.

Remark 4. The staining presented in Figure 3 are nice. However, how to reconcile those results with the ones obtained in figure 1? Again, what is significant and what is not?

Response 4. In Figure 3 (now Fig. 4) we observed clear differences in *C1GALT1* KO, *C1GALT1C1* KO and *MGAT1* KO, such as diminished glycoprotein surface presentation, or altered localization, which we illustrated with fluorescence intensity histograms and colocalization plots. This fits with findings identified as significant after the reanalysis of Figure 1, where diminished viral fitness was found for virus produced in these cell lines.

Remark 5. In Fig. 3C, how to explain the weaker signal at 20h pi than at 14h pi?

Response 5. Thank you for the comment. Since we are looking at fluorescent signal from the viral capsid protein, it is possible that at a later time point production of new capsids in the nucleus is either less intense or more dispersed. In Fig. 3c (now 4c), we included a later time point to probe, whether the infection is just delayed in *C1GALT1C1* KO. As we did not see more intense capsid production at 20 hpi, we concluded that the infection is overall less robust in the KO.

Action 5. We corrected the text on page 16 (lines 371-373) to clarify:

“In WT cells at 14 hpi, multiple capsid assembly sites could be seen in the nucleus and capsids were also associating with the nuclear envelope in most cells irrespective of the viral load (Figure 4c). In *C1GALT1C1* KO cells less and smaller assembly sites could be seen, and capsids were less frequently associating with nuclear envelope. **This association slightly improved later in infection (20 hpi), but the capsid production did not intensify, suggesting HSV-1 infection is generally less robust in *C1GALT1C1* KO (Fig. 4c).”**

Remark 6. In figure 4, the normalization relative to protein surface expression does not appear so robust. Indeed, surface expression signal depends on a large amount of proteins whereas, the success of fusion is revealed similarly when one or thousands of fusion events occurred per cell. This should be clarified. One solution would be to make the assay with dilutions of the same plasmid and then see if reduced cell surface expression translates in differences in fusion readout.

Response 6. Again, thank you for the comment, and we now reconsidered how to present the data. We should mention, that the assay has previously been extensively optimized at the Cohen-Eisenberg lab identifying gB as the rate limiting protein, where the amount of protein on the surface directly correlated with fusion efficiency (Atanasiu *et al.* J Virol 2013).

In our view it is a rather widespread practice to probe glycoprotein surface expression and correlate to fusion data to make sure the reduction in fusion is not related to diminished surface presentation (Lin and Spear PNAS 2007, Fan *et al.* mBio 2018). We agree, however, that upon moderate reduction in surface presentation, the success of fusion is not likely to be affected. We therefore removed the figures with surface normalization and instead presented percentages of fusion at a certain time point and percentages of surface expression in side by side columns (Figure 5g) for easier appreciation of these two different types of data.

Action 6. We updated Fig. 5

“ **f** Cell-cell fusion activity of gB mutants at t = 120 min. Data is shown as mean normalized luminescence from two independent experiments +SD. Two-way ANOVA followed by Dunnett’s multiple comparison test was used to evaluate differences from WT (* $p < 0.05$, ** $p < 0.01$, *** $p < 0.001$, **** $p < 0.0001$). **g** Average percentages of cell surface expression and fusion efficiency at t = 120 min from two independent experiments +SD are shown in side-by-side columns.”

Remark 7. What is the rationale to do it with gD in figure 4? As mentioned by the authors gD does not mediate fusion. However, the readout is fusion.

Response 7. Split luciferase assay is commonly used to evaluate the performance of all viral proteins needed for HSV-1 entry, and not only the fusion effector itself. Since fusion cannot happen without initial attachment via gD, we were anticipating that gross effects in binding efficiency would be reflected in fusion efficiency. We did not observe such differences using HEK293 cells as target, presumably expressing multiple receptors that can rescue binding. We therefore wanted to investigate this further and obtained CHO cell lines expressing Nectin 1 and HVEM separately. In conclusion, we saw a small reduction on Nectin 1-initiated fusion, and a greater reduction in HVEM-mediated fusion with compound mutations in gD.

Action 7. We performed experiments with CHO-HVEM and CHO-Nectin 1 as target, and included the data in the revised manuscript (Figure 5 l-o).

“l-o CHO cells stably expressing Nectin 1 (l, m) or HVEM (n, o) were used as target cells to evaluate cell-cell fusion activity using gD O-glycosite Ser/Thr to Ala mutants. l, n Cell-cell fusion activity over 180 min using CHO-Nectin 1 (l) or CHO-HVEM (n) as target. Data from two independent experiments is shown, where mean normalized luminescence of three technical replicates at each time point is indicated by a dot. Mean values of the two independent experiments are shown as thin lines. The parental CHO cell line without entry receptors was used as control and values were subtracted from all measurements. Data is normalized to maximum luminescence reading at final time point using WT gD for each experiment. m, o Cell-cell fusion activity of gD mutants at t = 120 min using CHO-Nectin 1 (l) or CHO-HVEM (n) as target. Data is shown as mean normalized luminescence from two independent experiments +SD. Two-way ANOVA followed by Dunnett’s multiple comparison test was used to evaluate differences from WT (* $p < 0.05$, ** $p < 0.01$, *** $p < 0.001$, **** $p < 0.0001$).”

Text addition on page 20 (lines 470-475):

“To inspect possible contributions of gD mutations to interactions with distinct HSV-1 entry receptors, we utilized CHO cells overexpressing Nectin 1 or HVEM as target (Fig. 5 l-o, Supplementary Fig. 3a, 3b). Here we saw a modest reduction in Nectin 1-initiated cell-cell fusion, when T255 and S260 were collectively mutated (Fig. 5l, 5m). A more pronounced reduction in cell-cell fusion efficiency was seen in HVEM-mediated entry upon introduction of these mutations (Fig. 5n, 5o).”

Remark 8. In Figure 5, when the authors suspect an effect of the mutations on the glycoproteins glycosylation (for example with gE N glycans), they could show it by western blot in order to support their hypotheses. This would be nice to see the molecular weight of gE in the different KO cells treated or not with specific enzymes removing glycans.

Response 8. This is a great suggestion. However, since gE only harbors 2 N-linked glycans, and since we only introduce subtle changes in N-glycan structure, we fear that the molecular weight difference would not be obvious on a simple Western Blot. Therefore, a more elaborate analysis is needed. Although this is beyond the scope of the present study, we anticipate this to be part of more detailed and future mechanistic studies.

Remark 9. The comparison of 2D with 3D cultures is very interesting, however, the results do not appear coherent. The technique applied to figure 5D-G is very nice, however the interpretation is difficult as, here too, most of the differences observed are not significant. This should be investigated further.

Response 9. We agree that many of the findings in the 3D cultures are more descriptive in nature. Still, we do identify a significant increase in lesion area in *MGAT1* KO, and the inability to form transepidermal lesions in *MGAT4A* and *MGAT5+4B* KO is evident, despite lack of significance in area difference. We therefore (again) agree that this should be investigated further. However, we believe this should be performed in a new and separate study given the enormous time and resources required to generate and infect the 3D cultures (4+ weeks) with subsequent sectioning, imaging, and analysis.

Minor comments

Remark 10. The legend of Figure 1 is not correct. It seems that there is a mix with Figures S1 and S2.

Action 10. Thank you for pointing out the error, which we corrected.

Remark 11. Before comparing viral growth on the KO cell lines, it could have been interesting to show that mutations do not affect cell survival or proliferation as it could have direct consequences on virus growth.

Response 11. This is a valid concern. However, we did not observe any major differences in the day to day passaging of the different knock outs, and all knock outs are able to form 3D cultures. We have previously evaluated the growth rate of *C1GALT1C1* KO cells, and it in fact proliferates slightly faster, yet exhibits diminished viral replication dynamics (Radhakrishnan *et al.* PNAS 2014, Bagdonaite *et al.* PLoS Pathog 2015).

Remark 12. Panels B-D of Figure 1 appear very interesting but they are always presented as ratio or ratio of ratio (panel D). It would have been interesting at some point to have an idea about the absolute number of PFU or viral genome copies.

Action 12. We changed the presentation of the data, as shown in revised Fig. 1 (see Response 1).

Remark 13. The decreased fitness suggested for virions produced on *MGAT1* KO cell seems to be non significant. The text should therefore be changed.

Action 13. The fitness is significantly decreased after the reanalysis.

Remark 14. The sentence in lines 165-166 is not clear. Indeed, how can it be said that binding is complete or not?

Action 14. We based the sentence on a previous publication, where all added virions bind after 3 h. As this is not essential information, we removed it.

Remark 15. Lines 389-390, could the author clarify the sentence: “where dissociation of progeny virions is restricted”

Action 15. This is related to reduced motility of progeny virions in semi-solid overlay media and thus their ability to spread beyond the initial local site of infection. We clarified the sentence on page 20 (lines 485-487):

“We first performed plaque assays with 2D grown cells, where dissociation of progeny virions is impeded by the dense overlay media, making direct cell-to-cell spread as the predominant mode of spread.”

Reviewer #2 (Remarks to the Author):

Remark 1. Authors are correct in stating that there are limited accessible research tools and model system to cellular and viral glycan biology. Within such context, glyco-engineered cells are welcome. However, with multiple different cell types in the human body, the 'right choice' of cells to have the glyco-engineering would attract the most attention from the community (in my opinion).

Response 1. We agree with the reviewer that the right choice of cells is important. For that reason, we glycoengineered a human keratinocyte cell line (HaCaT) derived from the human skin – the natural target organ and cell type that HSV-1 encounters at the oral skin surfaces. While skin cells obtained from the correct anatomical site would be even more relevant, using primary cells is problematic in terms of glycoengineering and subsequent assays due to eventual senescence. We thus believe our choice of cells is well balanced for ease of manipulation, yet relevant for investigating HSV-1 glycobiology in the natural-like context, and arguably of value for the community (see also general comment above).

Remark 2. Authors are also correct stating that glycans ‘being the most diverse post-translational modification’, glycan dynamically shaping the glycans on proteins and lipids through the secretory pathway. It is with these recognised challenges; authors perhaps need to control with their experiments much more tightly and / or tone down their claims. More specifically, the domino effect on any glycosylation related enzyme manipulation will impact on the target cells in terms of natural biology.

Specifically, any of these glycan modification in N-glycan or O-glycan will give rise to a series of domino effects that are not just related to viral proteins, but ALL cellular proteins rely on these specific glycosylation events. As authors pointed out, these wide spectra of glycosylated proteins will contribute to multiple series and layers of ripple effects on these viral and cellular glycol-conjugates to influence their biological processes. Consequently, virus titter and viral genome DNA copies numbers are blunt instruments to reveal specific

biological impacts. It would have been much nicer if the authors can have examined a specific biological process relates to HSV-1, go deeper into specific virological events.

Response 2. It is clear that manipulating glycosylation pathways may have not easy to predict effects on the cellular proteome, protein-protein interactions and signalling pathways. However, we argue that targeting select glycosyltransferases will have a more limited effect than e.g. chemically blocking whole pathways. Also, for example, our direct targeting of select branching points in the maturation of N-glycans ensures that we do not interfere with calreticulin and calnexin mediated effects on protein folding, which is often the problem when taking the targeted approach eliminating the entire N-glycan on specific protein sites. Moreover, a substantial portion of our data show no difference in viral behaviour upon knock out of specific enzymes. We acknowledge that we with this approach only scratch the surface of how glycans impact infection of HSV-1. Nonetheless, the strategy allows us to focus our efforts in future more mechanistic studies. We however agree that there is a need to better control for the impact of glycoengineering on cellular components important for the viral infection to further stratify the relevance of our findings, which we partially addressed in our revised manuscript.

After careful statistical reanalysis of all our data (see responses to Reviewer 1), we are more confident in our infectious cycle stage-specific findings, and edited the text to avoid speculation on non-significant observations. Furthermore, we have now made efforts in characterizing the cellular landscape of viral entry receptors Nectin-1 and HVEM in our knockout cell lines, partially explaining some of the effects we see in certain knock outs (Fig. 2i). Specifically, reduced levels of Nectin 1 may help explain reduced viral binding to *MGAT1* KO cells, whereas that is not the case for glycolipid knock out cells (*B4GALT5* KO and *ST3GAL5* KO). We thus further investigated the distributions of various glycolipid subtypes in human keratinocytes using antibodies and bacterial toxins (Fig 3). We found that the immunofluorescence staining patterns matched with the expected knock out glycophenotypes, further validating our cell lines. Furthermore, we found heterogeneous distribution of different subtypes of glycolipids across WT cell monolayers, as well as a polarized subcellular localization of more elaborate glycolipid structures, which may have bearings for the interaction with the extracellular virions.

Action 2. We performed new experiments and included the data in updated Figure 2 and new Figure 3.

Figure 2i

“ i Nectin 1 and HVEM surface expression in HaCaT cell lines was quantified by flow cytometry. Data points show background subtracted median fluorescence intensity (MFI)

from two independent experiments, and the bar heights indicate mean +SEM. MFIs of cells labelled with secondary antibodies only were used for subtracting background.”

Text additions on page 11 (lines 246-255):

“To follow up on our binding and entry data, we aimed to investigate the cellular landscape of HSV-1 entry receptors and other surface molecules that may have an impact on the early virus-cell interactions in the different knock out cells. We first quantified the surface expression levels of Nectin 1 and HVEM in WT HaCaT cells and found very low levels of the latter (Fig. 2i, Supplementary Fig. 3a, 3b). *MGAT1* KO and *B4GALT7* KO cells expressed significantly lower levels of Nectin 1 on the cell surface, whereas *MGAT4* KO and *GCNT1* KO expressed higher levels (Fig. 2i). These results correlate well with the virus binding data, and may help explain the altered proportion of virus bound to cells with alterations in N-glycosylation and O-glycosylation pathways. Importantly, the selective effect on entry to *MGAT4* KO was not correlated to availability of the receptor.”

“Figure 3. Heterogeneous expression of glycosphingolipids in HaCaT keratinocytes. a The cartoon depicts a simplified human glycosphingolipid biosynthetic pathway. Glycolipid

structures highlighted in magenta were probed by antibodies or fluorescently labelled toxins. **b-e** Cells grown on cover slips were fixed with 4 % PFA and stained for different GSL structures. All images are representative of 2 independent experiments. **b** Confocal micrographs show distribution of different GSLs in HaCaT WT, *B4GALT5* KO and *ST3GAL5* KO monolayers. Scale bars are indicated for each set of micrographs. **c** z-stack maximal intensity projection of HaCaT WT cells labelled with anti-GlcCer and anti-GM3 antibodies. Scale bar is indicated. **d** HaCaT WT cells labelled with anti-GM3 antibody. An individual z-slice within a stack is shown, with orthogonal cross sections of the z-volume included, and indicate apical expression of GM3. Nuclei are labelled with DAPI (blue). Scale bar is indicated. **e** The confocal micrograph shows spatially distinct distribution of Gb3 and GM3 GSLs in HaCaT WT, probed by FITC-labelled Shiga toxin B (StxB-FITC), and anti-GM3 antibody, respectively. Scale bar is indicated.”

Text additions on page 13 (lines 291-309):

“We next looked into GSLs expressed in skin cells (Fig. 3). We saw comparable levels of Nectin 1 on the surface of WT, *B4GALT5* KO and *ST3GAL5* KO cells (Fig. 2i), and yet HSV-1 binding and entry to these cells was markedly decreased. We thus hypothesized that elongated GSLs may help deliver the viral entry receptors to membrane compartments accessible to incoming virus. We used antibodies and toxins recognizing various (glyco)lipid structures to illuminate their distribution in keratinocytes (Fig. 3a). Ceramide and glucosylceramide, representing initial steps of GSL synthesis, were predominantly located intracellularly in WT cells, while some ceramide accumulation could be seen in *B4GALT5* KO, devoid of elaborate GSLs (Fig. 3b). Interestingly, expression of more complex GSLs was heterogeneous, and different cells appeared committed to a specific GSL subtype. Specifically, we detected Gb3 structures, synthesized from lactosylceramide precursor, in both WT, and *ST3GAL5* KO cells with clear surface presentation, but not *B4GALT5* KO (Fig. 3b, 3e). In contrast, GM3, the product of *ST3GAL5*, was only detected in WT cells (Fig. 3b). GM3 partially co-localized with intracellular glucosylceramide-positive structures but were primarily expressed on the cell membrane (Fig. 3c). Importantly, GM3 was abundantly found on apical cell surfaces accessible to the extracellular environment (Fig. 3d). Gb3 and GM3 were expressed in mostly distinct subsets of cells, and a substantial proportion of skin cells remained unlabelled, presumably expressing more elaborate structures (Fig. 3e). In conclusion, we show heterogeneous yet regulated expression of different GSLs in distinct cells and within different cellular compartments, which may be relevant for interaction with extracellular virus.”

Remark 3. The data outcome (claims) within the section in ‘Binding and entry HSV-1 to mutant cell lines’ are expected. In my opinion, one might have missed the opportunity to use these specific glyco-engineering to dissect the mechanisms better. In my opinion, current presentation might not be the best applications / usages of these really cool glycoengineered cells. With ALL glycan molecules on targeted cells are affected by a single glyco-related enzyme, it is extremely difficult to understand the precise biological impacts.

Response 3. It is important to distinguish the effects of non-redundant initiating enzymes changing the glycan structures on all the proteins, as rightfully pointed out by the reviewer, from those of branching and capping enzymes, which will only modify a proportion of glycans on a subset of proteins. While the outcomes may be expected for some, they have

not been suggested before. To our knowledge, requirement of glycosphingolipids for efficient HSV-1 binding and entry have not been described before, neither has the selective effect of MGAT4A on HSV-1 entry but not binding, or the relative contributions of GAG binding determinants clarified. We are aware that we do not present precise mechanisms, which we tried to emphasize in the manuscript. By electing to start with a broader screen, we do identify several important leads to follow up on in future studies.

Action 3. Based on data reanalysis (see “Responses 1 and 2” to Reviewer 1), we further took care not to overstate the impacts of our findings, while offering some additional insight based on the data on receptor characterization (“Response 2” above and “Response 5” to Reviewer 3).

Remark 4. It seems to me that the authors are much stronger glyco-biologists than virologists, and perhaps the pitch and the focus of the story should be presented as such.

In its current form, the manuscript appears more like to potpourri of some nice glycobiology observations with selected glycoengineered cells without a highly coordinated story for a high-impact journal of this calibrate. For example, using each one of these glyco-engineering cell to go deep and dissect a specific aspect of the story would be really nice.

Response 4. We acknowledge that this may often be perceived as a problem with larger and broad interdisciplinary studies. We attempted to undertake a systematic approach while using widely accepted HSV-1 life cycle-specific assays to identify classes and subtypes of glycans involved in these specific events. Together with the additional characterization of predominant entry receptors and distribution of glycolipids in human skin cells, we believe the paper to be both novel and comprehensive in its approach. Therefore, we believe that the study is a useful resource for the glycobiology and the virology community to investigate the involved pathways in detail.

Remark 5. The science is great, but the data appear to be disjoint (to me) in terms of storytelling. In current form, it feels more like a series of ‘capable experiments’ by a strong research team.

For example, while the platform of constructing these glycoengineered cells are highly valuable, but if the host cells are only with keratinocytes, the application with virology will be limited. Similarly, if the idea is to have glycoengineered cells to study biological processes, this assessor would like to see the platforms to be applied to other relevant primary cells that can accelerate biological and scientific discoveries in the context of relevant glyco-biological questions. Other mucosal cells or immune cells that rely on glycobiology to interact with pathogens would be excellent addition.

Response 5. In current work, we used a cell type most relevant for the lytic infection of HSV-1, related to the most common clinical manifestation of the virus at the primary site of infection in humans. The conclusions of current study will arguably allow to focus the efforts to investigate select glycosylation pathways in these difficult to engineer primary cell types, as well as pursue virological applications to limit viral spread in the skin (see also general comment above).

Remark 6. The domino effects of these glyco-engineering cells will always be a concern. Perhaps coupling with some mix-omics analyses on some of the relevant cell types would suit the purpose for platform for the field.

Response 6. Again, we agree that this will naturally be a concern. We have previously investigated the global transcriptomic profiles upon glycogene knock out of specifically *MGAT1*, *C1GALT1*, and *B4GALT5* in a different keratinocyte cell line, offering some insight into affected cellular pathways (Dabelsteen *et al.* Developmental Cell 2020). Clearly some of these will be relevant for viral infection, and others will not. In our previous work on O-glycan initiation in HaCaT cells, we identified limited proteomic changes, while phosphoproteomics and transcriptomics offered some insight into observed phenotypes, but was not sufficient to explain it (Bagdonaite *et al.* EMBO Reports 2020). Moving forward, correlating such data will be an important resource in dissecting the affected viral mechanisms in more detail, as it is now mentioned in the text.

Action 6. We included a section in the discussion reflecting on findings from our previous studies on page 28 (lines 694-699).

“It is important to consider the broader effects of glycogene knock out on cellular pathways and interaction networks. We have previously assessed transcriptomic profiles of *MGAT1* KO, *C1GALT1* KO, and *B4GALT5* KO N/TERT keratinocytes⁶, where diverse transcriptomic changes affecting cell signalling, adhesion, and differentiation were identified, some of which may have bearings for the viral infection. Moving forward, correlating such data will be an important resource in dissecting the mechanisms affecting viral infections in more detail.”

Remark 7. This assessor (personally) feel that there are many good parts in this manuscript that can be further developed to be high impacts paper on their own. In current form, it feels (to me) that the aspects being presented do not 'gel well' for this audience.

Response 7. We appreciate the reviewer’s point of view, and we also believe that there are a lot of interesting leads to follow up on in isolated studies. We, however, believe that the broader scope with a genetic entry point to probe the importance of glycans in the context of a viral infection should be of interest to a broader readership.

Reviewer #3 (Remarks to the Author):

Glycans are known to be exploited by multiple human pathogens. In this study authors reports the glycoengineering of HaCaT keratinocytes by introducing targeted deletions in the cellular glycome, including select initiation, branching and capping events, covering all the major classes of glycoconjugates to delineate human glycosylation pathways for their potential role for specific glycans in the infectious life cycle of herpes simplex virus type 1 (HSV-1). The study finds that GAGs and glycosphingolipids are important for HSV-1 attachment, while N-glycans play critical role for entry and spread, and O-glycans for needed for virus propagation. Further, it was found that the altered virion surface structures had minimal effects on the early interactions in the wild type cells, however the mutation of specific O-glycosylation sites affected glycoprotein surface expression and function. In

conclusion, the study highlights the significance of specific glycans in a clinically relevant human model of HSV-1 infection and urges the further exploration of genetic engineering may further to elucidate novel antiviral interventions.

Certainly, the work is novel and has significance in the herpes entry field. The experiments and methodology along with the controls have been done carefully and manuscript should be considered for publication after clarification with some additional experiments for evidence to further strengthen the overall goals of this finding.

Remark 1. Since the glycans are known regulators as a house keeping gene, it is obvious to ask if the KO Keratinocytes library affected membrane topography with the GAG expression and potential cellular signaling ability. An assessment profiles for the above compared to control will be useful.

Response 1. It is true that knocking out glycosyltransferases will potentially affect the expression and function of many different proteins and cell signaling pathways (see also “Response 6” to Reviewer 2). It is not clear, whether eliminating enzymes in one glycosylation pathway would affect the levels of enzymes in other glycosylation pathways. We have previously seen limited changes in RNA expression profiles of other glycosyltransferases in human N/TERT keratinocytes with knock out of *MGAT1* or *C1GALT1/C1GALT1C1* or *B4GALT5* KO (Dabelsteen *et al.* Developmental Cell 2020). However, it is clear that eliminating all GAG chains upon knockout of *B4GALT7* should have a meaningful impact on the apparent glycoalyx. While we lack good methods to visualize intact GAGs on the cell surface, we can release the GAGs from the cells and quantify the composition of disaccharides the GAGs are made of.

Action 1. We performed disaccharide analysis on wild type and *B4GALT7* KO keratinocytes (Fig 2 j, k, Supplementary Fig. 4), and found out that both chondroitin sulfate and heparan sulfate type GAGs were eliminated. However, hyaluronan, synthesized by non-competing enzymes was still detected. Furthermore, we found decreased expression of HSV-1 entry receptor Nectin 1 (Fig 2i) on the surface of *B4GALT7* KO cells, further limiting the availability of molecules for the virus to interact with.

“j The bar graph shows percentages of total quantified CS disaccharides in HaCaT WT from Supplementary Table 2. k The bar graph shows percentages of total quantified HS disaccharides from Supplementary Table 3.”

Supplementary Figure 4. Disaccharide analysis of HaCaT WT and *B4GALT7* KO cells.
a Chondroitin sulfate (CS) analysis. **b** Heparan sulfate (HS) analysis.

Text edits on page 11 (lines 264-267) and page 12 (lines 271-275):

“Except for hyaluronan, which is synthesized by a distinct family of enzymes, we did not detect any CS or HS disaccharides in *B4GALT7* KO cells (Supplementary Fig. 4). HaCaT WT cells expressed high levels of 4-O-sulfated or 6-O-sulfated CS, hyaluronan, as well as N-sulfated, N-/2-O-sulfated, N-/2-O/6-O-sulfated, and non-sulfated HS.”

“The disaccharide expression profiles in skin cells provided additional insight into the gC binding data on the CHO cell library. Namely, N-sulfated GAG motifs required for gC binding to CHO cells were abundantly found on human keratinocytes, and likely play a significant role *in vivo*. On the contrary, 4-O-sulfated CS, abundantly found on skin cells, is unlikely to be a critical receptor for gC, as seen from CHO data.”

Remark 2. In fact, a control human herpesvirus (CMV or EBV) should have been used to reach to the conclusion (but it is not required).

Response 2. It would indeed be curious to explore effects of glycosylation on other human herpesviruses. We would not, however, expect identical results to HSV-1, as beta- and gammaherpesviruses utilize other entry receptors and preferentially infect other cell types than keratinocytes in the human body. Understanding the impact of glycosylation for these viruses would without doubt make an informative follow up study.

Remark 3. In addition, multiple HSV-1 gD receptors are also tethered with the cell membrane, it is critical to evaluate if KO Keratinocytes (mutants) may have also affected their expression (nectin/HVEM) level. This can be shown either by cell-ELISA or flow cytometry.

Response 3. Thank you for this good suggestion. We agree that altering cellular glycosylation pathways may directly or indirectly affect the expression levels of HSV-1 entry receptors, which should be investigated.

We performed flow cytometry analysis on all the mutant cell lines using Nectin 1 and HVEM specific antibodies (see also “Response 2” to Reviewer 2). We found decreased levels of Nectin 1 in cell lines lacking *MGAT1* and *B4GALT7*, and increased levels in *MGAT4A* KO and *GCNT1* KO (Fig. 2i). This helps explain the lower binding capacity to *MGAT1* KO, and increased binding to *GCNT1* KO. For *B4GALT7* KO, we believe that the roughly 60 % lower levels of Nectin-1 would not explain the complete loss of binding, which is rather related to lack of early virus-GAG interactions. We detected very low HVEM expression in the HaCaT cell lines, in contrast to a CHO cell line overexpressing HVEM (Supplementary Fig. 3).

Action 3. We included additional data in the manuscript
Figure 2

“ i Nectin 1 and HVEM surface expression in HaCaT cell lines was quantified by flow cytometry. Data points show background subtracted median fluorescence intensity (MFI) from two independent experiments, and the bar heights indicate mean +SEM. MFIs of cells labelled with secondary antibodies only were used for subtracting background.”

Supplementary Fig. 3

Remark 4. The intriguing question remain unanswered that which gD receptors and internalization modes along with glycans mediate the HSV entry in vivo at the natural sites of virus infection.

Response 4. Though we did not look into internalization modes, we believe our additional data helps answer the question, which gD receptors are utilized in human keratinocytes. We found moderate levels of Nectin 1, and low levels of HVEM and 3-O-sulfated HS (Fig 2i-k, see also “Response 5”). A recent study has also confirmed expression of Nectin 1 in N/TERT keratinocytes, and shown that CRISPR/Cas9 KO of Nectin 1 severely diminished HSV-1 entry (Kite *et al.* PLoS Pathog 2021), which supports our findings.

Remark 5. It is unclear if the sulfated GAGs contribute only to viral attachment (as per the comments made in the manuscript), since 3-O sulfated-HS has been shown to generate as an independent gD receptor allowing virus-cell fusion.

Response 5. Thank you for pointing this out. While the gD-3-O-sulfated-HS interaction has long been established, initially based on overexpression of 3-O sulfotransferases and inhibition of entry by soluble 3-O HS (Tiwari *et al.*, J Gen Virol 2004), it remains unclear, whether this receptor is utilized at the site of infection due to analytical difficulties in detecting differential sulfation patterns on GAGs. To answer this question, we performed disaccharide analysis of released cellular GAGs (see also “Response 1”), and included our recently chemoenzymatically synthesized 3-O sulfated HS disaccharide standards (Karlsson *et al.* Science Advances 2021). Our data shows that 3-O sulfated GAGs comprise only minute amounts of the total HS, and thus it is unlikely that they have a large contribution to viral entry to keratinocytes (Fig 2k, Supplementary Fig. 4).

Action 5. We added the following text on page 11 (lines 256-270):

“For *B4GALT7* KO, Nectin 1 presentation decreased by approximately 60 %, but this does not explain the complete loss of HSV-1 binding, which is likely a combination of a decrease in glycosaminoglycan and protein receptors. While gC mediates early virus-GAG interactions, facilitating subsequent interactions between gD and its cognate protein entry receptors, 3-O-sulfated HS has also been identified as an independent entry receptor for gD³⁴. In order to evaluate the potential contribution of 3-O-sulfated HS to HSV-1 entry in skin cells, we

performed disaccharide analysis of HaCaT WT and *B4GALT7* KO cells, which included our recently synthesized 3-O-sulfated HS disaccharide standards (Fig. 2 j, k, Supplementary Fig. 4, Supplementary Table 2 and 3). Except for hyaluronan, which is synthesized by a distinct family of enzymes, we did not detect any CS or HS disaccharides in *B4GALT7* KO cells (Supplementary Fig. 4). HaCaT WT cells expressed high levels of 4-O-sulfated or 6-O-sulfated CS, hyaluronan, as well as N-sulfated, N-/2-O-sulfated, N-/2-O/6-O-sulfated, and non-sulfated HS. We detected very low levels of 3-O-sulfated HS disaccharides, demonstrating that usage of these receptors for HSV-1 entry in human keratinocytes is unlikely. We therefore suggest that Nectin 1 is the most widely available HSV-1 entry receptor for gD in HaCaT keratinocytes.”

Remark 6. 200 pfu needs to be expressed in terms of MOI, as some places the infections are represented as MOI.

Response 6. In those instances, we were adding a fixed number of PFU to confluent cell monolayers with slightly different total cell numbers to generate isolated plaques, where only one virion will lead to the development of a lesion with lesions spaced far away from each other. Therefore, we cannot express the PFU as a precise MOI. We can estimate that the MOI will likely be < 0.0005.

Action 6. We added “MOI < 0.0005” to relevant figure legends and Materials and Methods.

Remark 7. The Fig. 1E, and 1F mentioned in the figure legends are missing.

Action 7. Thank you for pointing out the mistake, we corrected the errors in Figure 1 legend labels.

Remark 8. Figure 4 B cell to cell fusion model is not well explained, why the presence of cell surface glycans in CHO and HEK293T cells have been ignored. What will be the type(s) of gD receptor in HEK293T cells?

Response 8. The CHO cells have been described to be refractory to HSV-1 infection as they do not express any entry receptors for HSV-1, and this is the reason why they are used as effector cells. Since there are no entry receptors, expression of the HSV-1 fusion machinery will not induce syncytium formation in these cells. We chose HEK293 cells as a target cell because they are easy to transfect and are readily infected with HSV-1. Regarding the glycosylation status of the two types of cells, they utilize typical mammalian glycosylation pathways, and glycoprofiling data of these cell types are available (Yang *et al.* Nat Biotechnol 2015, Narimatsu *et al.* Molecular Cell 2019). Since we are only eliminating specific O-glycosylation acceptor sites on gB, the O-glycan structures at the remaining glycosites will be unchanged, and comparable in all our sample groups. Likewise, the glycoprofiles of target cells are the same in all compared groups.

We included HEK293 cells in our flow cytometry analysis for Nectin 1 and HVEM. Consistently with publically available RNA_Seq data, we found low levels of Nectin 1 and HVEM (Supplementary Fig. 3c), thus it must be a different gD entry receptor that is being

utilized, or entry is supported by gB co-receptors such as NMHC-IIA, which is highly expressed in HEK293 cells. Furthermore, to evaluate the contributions of gD mutations to utilize the classical HSV-1 entry receptors, we obtained CHO cells stably expressing Nectin 1 or HVEM, and performed cell-cell fusion assays using these cells as a target. We found that gD mutations affected the interaction with individual receptors. We observed reduced cell-cell fusion efficiency with compound mutations in gD, when CHO-HVEM were used as target, which was less pronounced when using CHO-Nectin 1 cells as a target (Fig. 5 l-o) (see “Response 7” to Reviewer 1).

Action 8. We explained the assay in the text better and performed flow cytometry analysis of HSV-1 entry receptors on HEK293 cells.

Text additions on page 19 (lines 449-452):

“We used CHO cells, refractory to HSV-1 entry, as effector, and HEK293, an HSV-1 permissive epithelial cell line, as target. We quantified low levels of Nectin 1 and HVEM on HEK293 cells (Supplementary Fig. 3c), suggesting other types of receptors and co-receptors may also be involved.”

Supplementary Fig. 3c

“c Nectin 1 and HVEM cell surface expression in HEK293 cells. Background corrected median fluorescence intensity from two independent experiments is shown.”

Remark 9. Figure 4 D, E and H have two panels and therefore should be re-numbered.

Action 9. We made sure all panels are appropriately numbered in the updated figure.

Remark 10. Fig. 4 C why gB expression itself is affected? Please provide the stats.

Response 10. There are several possible reasons for the changes in surface expression. Linker insertion mutagenesis studies has previously identified amino acid mutations which affect gB cell surface expression, most likely due to conformational changes induced in the protein. Another explanation could be, that glycosylation itself is important for secretion or stability. O-glycosylation has previously been suggested to regulate the secretory pathway transit time and Golgi exit (Sun *et al.* JBC 2020).

Action 10. We performed statistical analysis of the surface expression data and updated the figure (Fig. 5c).

“c Cell surface expression of gB O-glycosite Thr to Ala mutants evaluated by CELISA using mouse anti-gB antibodies. Data is shown as mean absorbance at 450 nm +SD of three technical replicates and is representative of three independent experiments. One-way ANOVA followed by Dunnett’s multiple comparison test was used to evaluate differences from WT (* $p < 0.05$, ** $p < 0.01$, *** $p < 0.001$, **** $p < 0.0001$).”

Remark 11. Please fix the Fig. 5C, why upper (higher x) and lower panels (lower x) are not taken from the same image.

Response 11. The images were taken from different fields with two different magnifications using a confocal microscope. The upper panel images were acquired with a 10x air objective and a tiling feature, stitching 4 images encompassing the area with a plaque. The lower panel images were acquired with a 63x immersion objective.

Action 11. We clarified in the legend that the images were acquired separately.

“c Cell monolayers grown on cover slips were infected with 200 PFU (MOI < 0.0005) of HSV-1 K26-GFP and overlaid with semi-solid media for 48 h followed by fixation and staining for HSV-1 gE (magenta). GFP labelled capsid proteins (VP26) are seen in green. Confocal images at two different magnifications were taken to illustrate overviews of plaques (4 combined tiles at 10x, upper panels) as well as gE expression at higher resolution (63x, lower panels). Scale bars for the different magnifications are indicated.”

Remark 12. Figure 5F, please mention green stands for HSVP26 GFP and blue stains for what?

Action 12. We clarified in the legend that blue corresponds to DAPI stain of the nuclei.

Remark 13. Discussion section says – “We utilized a genetically engineered keratinocyte library to systematically investigate the contributions of different classes of glycan structures to multiple stages of HSV-1 life cycle in the natural context of infection”. Please be specific and clear that here you are talking about cellular glycans but not the viral glycans. Are viral and cellular glycans can have homotypic or heterotypic interactions as well.

Response 13. To be precise, depending on the assay, we do address roles of either cellular or the viral glycans. Because the virus will acquire host-like glycans when replicating in the cell, e.g. lower viral titers will be a reflection on either cellular or viral glycans, or both; the same can be said on plaque size development, where after replication in the first cell, an engineered virus will spread in engineered cells. Only in binding/entry experiments either using wild type virus on engineered cells, or engineered virus on wild type cells, can we conclusively say that it is only the cellular and the viral glycans involved, respectively. This we have now clarified further in the text.

Action 13. We clarified the sentence on page 23 (lines 542-544).

“We utilized a genetically engineered keratinocyte library to systematically investigate the contributions of different classes of cellular glycan structures to multiple stages of HSV-1 life cycle in the natural context of infection and also investigated the properties of glycoengineered virions.”

Remark 14. Discussion section – “Since distinct glycan classes could independently finetune early virus-host interactions, we suggest that HSV-1 through its evolution developed elaborated mechanisms to exploit the host glycosylation machinery”.- what is basis of such statement, did you check the evolutionary herpes viruses that potentially affected aquatic species and had differences in terms of glycan selection?

Response 14. This is an excellent remark. One of the theories in glycobiology is that the diversity of vertebrate glycans is a result of evolutionary battle between pathogens and their hosts. It would indeed be very interesting to investigate contributions of cellular glycans in different animal species prone to herpesvirus infections. Based on what we know from the literature, fish, being vertebrate, will have fairly similar cellular glycans to humans. Mollusks also have similar overall classes of glycans, though different terminal structures decorating those glycans.

Action 14. We included a reference to a paper discussing evolutionary aspects of glycosylation to the sentence on page 23 (line 552).

Remark 15. In Discussion section – “We identified diverse effects on the infectious cycle attributable to select maturation or branching steps of protein-bound N- and O-linked glycans” – again be specific that here you are highlighting the significance of viral glycans.

Response 15. Again, in the context of the plaque size, discussed further in the section, we cannot unequivocally state, whether it is the viral or the cellular glycans (see also “Response 13”).

Action 15. We added additional text in discussion on page 28 lines (683-685):

“It should be emphasized that both the cells and the spreading virus are glycoengineered in such setups, and pinpointing the viral or the host protein responsible for the observed effects requires more detailed studies.”

Remark 16. Does all the clinical HSV strains presumably loaded with variable viral glycans could use similar cell glycan signature to enter the host cell, or the phenomenon could just

be strain specific/cell specific is not clear. The clinical significance of glycoengineered viruses is not clear. It would have been great to use clinical isolates in this study.

Response 16. We can only speculate that the overall glycoprofiles of different virus isolates would be similar, despite having derived from circulation in different humans, and it is unlikely that the minor potential differences in relative abundances of the subtypes of viral glycans would make a difference in their interactions with the host. However, there might be differences in site-specific glycosylation. We have previously analyzed a laboratory strain and a clinical isolate of varicella zoster virus, another alphaherpesvirus (Bagdonaite *et al.* JBC 2016), where we found overall similar site-specific O-glycosylation patterns, but also some notable differences. The potential impact on interaction with host cells would need to be evaluated on a case-to-case basis.

Remark 17. Since the glycans are also involved both in the interaction of a prokaryotic bacterium or virus binding to a eukaryotic host cell, and the interactions between the cells of the immune system, nothing has been discussed in terms of nosocomial infection which could potentially utilize similar route. Therefore, the overall implications or significance of the current study may have propounding impact in the future.

Response 17. We agree that glycoengineered cell platforms of relevant cell types can be used in probing the interplay of various pathogens with their hosts.

Action 17. We added a sentence in the discussion highlighting broader applications (page 28 (lines 705-706)).

“We furthermore highlight a glycoengineered human cell platform as a versatile tool for illuminating the glycobiology of other viral, bacterial, and fungal microorganisms.”

Remark 18. Furthermore, Glycans have been shown to function as versatile molecular signals in cells and therefore it is not clear if blocking or targeting glycans would bring an unwanted effect on the host cell.

Response 18. We do have some knowledge on transcriptional and signaling changes in cells with certain glycosyltransferase knock out (Radhakrishnan *et al.* PNAS 2014, Bagdonaite *et al.* EMBO Reports 2020, Dabelsteen *et al.* Developmental Cell 2020). Not all of those changes may have relevance for a viral infectious cycle, but are worth considering when interpreting data. It is clearly one of the drawbacks of a genetic entry point.

Remark 19. Lot of references are missing in the original statement throughout the manuscript.

Response 19. We regret having missed out on some important literature, though it quite difficult to pinpoint, which statements the reviewer is referring to. We will gladly add the missing references, if the reviewer provides some more specific details.

REVIEWERS' COMMENTS

Reviewer #1 (Remarks to the Author):

The authors have taken into account the feedback provided by the reviewers, resulting in a noteworthy enhancement of the manuscript. While the manuscript has been substantially improved, questions persist regarding the biological significance of the findings. The presented data largely establish correlations rather than mechanistic connections between glycosylation differences and specific aspects of the virus lifecycle, even if some parts related to entry are addressed. Nevertheless, the innovative methodology employed and the overall quality of the research render the data compelling and relevant to the field.

Reviewer #2 (Remarks to the Author):

I am happy with the efforts made by the authors to revise the manuscript that have addressed my scientific concerns.

Response to referees

Reviewer #1 (Remarks to the Author):

The authors have taken into account the feedback provided by the reviewers, resulting in a noteworthy enhancement of the manuscript. While the manuscript has been substantially improved, questions persist regarding the biological significance of the findings. The presented data largely establish correlations rather than mechanistic connections between glycosylation differences and specific aspects of the virus lifecycle, even if some parts related to entry are addressed. Nevertheless, the innovative methodology employed and the overall quality of the research render the data compelling and relevant to the field.

Response

We appreciate the positive feedback on the revised manuscript. We hope to investigate the most promising findings in more detail in future work.

Reviewer #2 (Remarks to the Author):

I am happy with the efforts made by the authors to revise the manuscript that have addressed my scientific concerns.

Response

We are glad to hear our efforts to revise the manuscript were satisfactory.